



Atmospheric
Measurement
Techniques
# The effect of rapid relative-humidity changes on fast filter-based aerosol-particle light-absorption measurements: uncertainties and correction schemes

**Sebastian Düsing**[1], **Birgit Wehner**[1], **Thomas Müller**[1], **Almond Stöcker**[2], and **Alfred Wiedensohler**[1]

[1]Leibniz Institute for Tropospheric Research (TROPOS), 04318 Leipzig, Germany
[2]Department of Statistics, Ludwig-Maximilians-Universität München CE1, 80539 Munich, Germany

**Correspondence:** Sebastian Düsing (duesing@tropos.de)

**Abstract.** TS1 Measuring vertical profiles of the particle light-absorption coefficient by using absorption photometers may face the challenge of fast changes in relative humidity (RH). These absorption photometers determine the particle light-absorption coefficient due to a change in light attenuation through a particle-loaded filter. The filter material, however, takes up or releases water with changing relative humidity (RH in %), thus influencing the light attenuation.

A sophisticated set of laboratory experiments was therefore conducted to investigate the effect of fast RH changes ($\mathrm{dRH}/\mathrm{d}t$ CE2) on the particle light-absorption coefficient ($\sigma_{\mathrm{abs}}$ in Mm$^{-1}$) derived with two absorption photometers. The RH dependency was examined based on different filter types and filter loadings with respect to loading material and loading areal density. The Single Channel Tricolor Absorption Photometer (STAP) relies on quartz-fiber filter, and the microAeth® MA200 is based on a polytetrafluoroethylene (PTFE) filter band. Furthermore, three cases were investigated: clean filters, filters loaded with black carbon (BC), and filters loaded with ammonium sulfate. The filter loading CE3 areal densities ($\rho^*$) ranged from 3.1 to 99.6 mg m$^{-2}$ in the case of the STAP and ammonium sulfate and 1.2 to 37.6 mg m$^{-2}$ in the case the MA200. Investigating BC loaded CE4 cases, $\rho^*_{\mathrm{BC}}$ was in the range of 2.9 to 43.0 and 1.1 to 16.3 mg m$^{-2}$ for the STAP and MA200, respectively. In addition, the effect of a silica-bead-based diffusion on the RH effect was investigated.

Both instruments revealed opposing responses to relative-humidity changes ($\Delta$RH) with different magnitudes. The STAP shows a linear dependence on relative-humidity changes. The MA200 is characterized by a distinct exponen-

tial recovery after its filter was exposed to relative-humidity changes. At a wavelength of 624 nm and for the default 60 s running average output, the STAP reveals an absolute change in $\sigma_{\mathrm{abs}}$ per absolute change of RH ($\Delta\sigma_{\mathrm{abs}}/\Delta$RH) of 0.14 Mm$^{-1}$ %$^{-1}$ in the clean case, 0.29 Mm$^{-1}$ %$^{-1}$ in the case of BC loaded filters, and 0.21 Mm$^{-1}$ %$^{-1}$ in the case filters loaded with ammonium sulfate. The 60 s running average of the particle light-absorption coefficient at 625 nm measured with the MA200 revealed a response of around $-0.4$ Mm$^{-1}$ %$^{-1}$ for all three cases. Whereas the response of the STAP varies over the different loading materials, in contrast, the MA200 was quite stable. The response was, for the STAP, in the range of 0.17 to 0.24 Mm$^{-1}$ %$^{-1}$ and, in the case of ammonium sulfate loading and in the BC loaded case, 0.17 to 0.62 Mm$^{-1}$ %$^{-1}$. In the ammonium sulfate case, the minimum response shown by the MA200 was $-0.42$ with a maximum of $-0.36$ Mm$^{-1}$ %$^{-1}$ and a minimum of $-0.42$ and maximum $-0.37$ Mm$^{-1}$ %$^{-1}$ in the case of BC CE5. Using the aerosol dryer upstream, the STAP did not change the behavior, but the amplitude of the observed effect was reduced by a factor of up to 3.

A linear correction function for the STAP was developed here. It is provided by correlating 1 Hz resolved recalculated particle light-absorption coefficients and RH-change rates. The linear response is estimated at CE6 10.08 Mm$^{-1}$ s$^{-1}$ %$^{-1}$. A correction approach for the MA200 is also provided; however, the behavior of the MA200 is more complex. Further research and multi-instrument measurements have to be conducted to fully understand the underlying processes, since the correction approach resulted in different correction parameters across various experiments. However, the exponen-

**Published by Copernicus Publications on behalf of the European Geosciences Union.**

tial recovery after the filter of the MA200 experienced a RH change could be reproduced. However, the given correction approach has to be estimated with other RH sensors as well, since each sensor has a different response time. And, for the given correction approaches, the uncertainties could not be estimated, which was mainly due to the response time of the RH sensor. Therefore, we do not recommend using the given approaches. But they point in the right direction, and despite the imperfections, they are useful for at least estimating the measurement uncertainties due to relative-humidity changes.

Due to our findings, we recommend using an aerosol dryer upstream of absorption photometers to reduce the RH effect significantly. Furthermore, when absorption photometers are used in vertical measurements, the ascending or descending speed through layers of large relative-humidity gradients has to be low to minimize the observed RH effect. But this is simply not possible in some scenarios, especially in unmixed layers or clouds. Additionally, recording the RH of the sample stream allows correcting for the bias during postprocessing of the data. This data correction leads to reasonable results, according to the given example in this study.

## 1  Introduction

Black carbon (BC) and its light-absorbing properties has significant influence on the Earth's climate, and its contribution is associated with major uncertainties, in particular due to its vertical distribution (Zarzycki and Bond, 2010). In addition, it is suspected to affect human health (WHO, 2012). Absorption photometers are instruments capable of measuring CE7 the light-absorbing properties of aerosol particles. These photometers measure the aerosol-particle light-absorption coefficient ($\sigma_{abs}$) by detecting the change of attenuation of light due to deposited aerosol-particle mass on sample filter. They have been installed on airship platforms (Rosati et al., 2016), tethered balloon platforms (Ran et al., 2016; Ferrero et al., 2014, 2016), or unmanned aircraft systems (UASs; Markowitz TS2 et al., 2017; Telg et al., 2017; Bärfuss et al., 2018) to address the vertical BC distribution. To investigate human exposure to health-harming BC-containing aerosol particles from combustion sources, they have been used for mobile measurements (Capeda TS3 et al., 2017, and references therein; Alas et al., 2018).

Subramanian et al. (2007), Vecchi et al. (2013), and Lack et al. (2008) have shown that liquid-like brown carbon can significantly bias filter-based absorption measurements, since this organic carbon wraps around filter fibers and alters their structural properties. Aerosol samples contain water vapor represented by its relative humidity (RH). Similar to liquid-like brown carbons, water vapor can be adsorbed through the filter material or bound to the binding material within the filter during the sampling process. A variety of filter materials are used in absorption photometers,

and the water uptake is different across various materials. Hence, changes in the aerosol RH can affect the aerosol-particle light-absorption measurements differently. Nessler et al. (2006) have shown the extent to which sudden changes in relative humidity (RH) can influence measurements of a Particle/Soot Absorption Photometer CE8 (PSAP; Radiance Research, Seattle, WA) and an aethalometer for clean filter material and that loaded with BC, whereas Cai et al. (2014) have shown the effect for the microAeth® AE51, although they did not quantify it. However, hygroscopic aerosol-particle species such as ammonium sulfate can take up water depending on the relative humidity. The RH effect for filters loaded with such aerosol species was never quantified. Furthermore, not all filter materials have been covered within these studies. In summary, both the filter material and loading material may influence the light attenuation of the filter.

The RH effect might not be relevant for averaging periods longer than 5 min, as usually done at stationary measurements on the ground (e.g., to address human exposure to BC-containing aerosol particles). However, to address vertical profiling of BC with fast RH changes, particle light-absorption measurements require a high temporal resolution of about CE9 seconds.

Telg et al. (2017) presented a study using an unmanned aircraft system (UAS) for vertical profiling of aerosol physical properties, including the aerosol-particle light-absorption coefficient measured by an absorption photometer. In their study, a significant decrease in $\sigma_{abs}$ at around 1000 m altitude is visible. Considering the other simultaneously measured microphysical aerosol parameters, this decrease is not to be expected. In the *WMO/GAW Report No. 227* (WMO/GMO, 2016), it is recommended to conduct aerosol sampling below 40 % relative humidity to prevent measurement artifacts due to high relative humidity. Although the measurements of Telg et al. (2017) have been conducted following these recommendations, this is a published example for the bias in $\sigma_{abs}$ measurements due to fast RH changes even for a RH below the 40 % threshold.

In our study, the RH effect is investigated for the small-sized photometers, namely the Single Channel Tricolor Absorption Photometer (STAP), using a quartz-fiber glass filter, and MA200®, which relies on a polytetrafluoroethylene (PTFE) filter. We show results of a set of laboratory experiments and address the effect of sudden changes in relative humidity for both absorption photometers. Herein, we consider three different scenarios: (a) clean filters, different filter loading densities of (b) hydrophobic BC, and (c) hydrophilic ammonium sulfate (($NH_4$)$_2$$SO_4$). In all of these cases we also investigated the impact of a silica-bead-based diffusion drier to the RH effect.

The following scientific questions are addressed: to what extent are STAP and MA200 sensitive to RH changes, and does different loading with respect to material and areal density contribute to this effect? Can the observed effect be corrected, and which recommendations can be given for

the usage of such an absorption photometer? This is important because recent developments indicate that lightweight absorption-measuring instruments will be used more frequently for airborne applications in the near future.

## 2   Experiment

### 2.1   Theory of absorption measurements

Filter-based absorption photometer measuring the decrease in intensity of light which passes through the filter medium with a specific optical thickness. The decrease in intensity can be described quantitatively according to the law of Beer–Lambert:

$$\ln(I) = \ln(I_0) + \sigma_{\mathrm{ATN}}(\lambda)l, \tag{1}$$

where $I$ is the attenuated intensity of light with a wavelength $\lambda$ with a raw intensity $I_0$, attenuated along a path $l$ through a medium with a light attenuation coefficient $\sigma_{\mathrm{ATN}}$. The path length $l$ can also be interpreted as the length of a column of aerosol passing through the sample area of the filter spot $A_i$ of the instrument (subscript $i$), whereas the particles are collected and accumulated in the filter. However, a reinterpretation of the path length does not mean that the result is the particle light-absorption coefficient but still the light attenuation coefficient. The path length $l$ can be calculated by the volume, which flows at a certain rate (volume flow rate; $Q_i$) for a time $\Delta t$ through the sample area $A_i$. Based on Eq. (1), this results in

$$\ln(I(t)) = \ln(I(t - \Delta t)) + \sigma_{\mathrm{ATN},i}(\lambda) \frac{Q_i \Delta t}{A_i}. \tag{2}$$

While aerosol particles deposit on the filter, the incoming light gets additionally scattered by those particles. Hence the effective pathway of the light through the filter increases due to the multiple scattering. To account for this, Eq. (2) needs $f(\tau)$, a transmission-dependent ($\tau$-dependent) filter-loading correction factor, with

$$\tau = \frac{I(t)}{I_0}, \tag{3}$$

where $I_0$ is the light intensity measured for a white, clean filter. For instance, Ogren (2010) reported an updated loading correction function for the PSAP introduced and updated by Bond et al. (1999), defined as

$$f(\tau) = (1.0796\tau + 0.71)^{-1}, \tag{4}$$

which is also used for the STAP.

Rearranging and applying the filter loading correction to the attenuation coefficient Eq. (2) will give the particle light-absorption coefficient:

$$\sigma_{\mathrm{abs},i}(\lambda) = \ln\left(\frac{I(t)}{I(t - \Delta t)}\right) \frac{f(\tau)A_i}{Q_i \Delta t}. \tag{5}$$

Water has a refractive index of $1.33 + i1.5 \times 10^{-9}$ at the 532 nm wavelength. Hence it interacts with incoming electromagnetic radiation. If the filter is exposed to relative-humidity changes, the light attenuation of the filter changes, since the water binds to the filter itself (Caroll, 1976, 1986). Since a variety of filter materials with different physical properties exist, we suspect that the magnitude and sign of the light attenuation coefficient can vary with the filter material. The hypothesis is that the change rate of the RH ($\mathrm{dRH}/\mathrm{d}t$) directly determines the magnitude of the particle light-absorption coefficient, which depends on the difference of two subsequent attenuation measurements.

The effect of changes in the relative humidity on particle light-absorption measurements contains two parts. First, the filter interacts with the RH on its own. Second, hygroscopic particles change their optical properties with the water uptake and loss due to growing and shrinking. Hence, aerosol particles under varying relative conditions will also have an effect on the reported particle light-absorption coefficient.

Some absorption photometers such as the STAP directly report measurements of the aerosol-particle light-absorption coefficient $\sigma_{\mathrm{abs}}$; some, for instance the MA200, report measurements of the equivalent black-carbon (eBC; Petzold et al., 2013) mass concentration ($M_{\mathrm{eBC}}$). eBC mass concentrations can be converted to $\sigma_{\mathrm{abs}}$ with

$$\sigma_{\mathrm{abs}} = M_{\mathrm{eBC}} \cdot \mathrm{MAC}, \tag{6}$$

in which MAC is the mass absorption cross section (in $\mathrm{m}^2\,\mathrm{g}^{-1}$).

Lab comparison of the eBC mass concentration between a MAAP (Multi Angle Absorption Photometer; Thermo Fisher Scientific, 27 Forge Parkway, 02038 Franklin, MA, USA; Petzold and Schönlinner, 2004) at the 637 nm wavelength and a MA200 at the 625 nm wavelength and the STAP at the 624 nm wavelength before the experiment revealed a good agreement, within 3 % and within 6 %, respectively. For the STAP, a MAC of $6.6\,\mathrm{m}^2\,\mathrm{g}^{-1}$ was assumed. Since a MAC of $6.6\,\mathrm{m}^2\,\mathrm{g}^{-1}$ is used for the MAAP at 637 nm, in this study we used the $\sigma_{\mathrm{abs}}$ directly provided by the STAP and derived with the mentioned MAC in the case of the MA200, which already accounts for multiple scattering and filter loading corrections.

### 2.2   Instrument description

As mentioned before, we investigated two filter-based absorption photometers, which are described in the upcoming sections. The STAP (Brechtel Manufacturing Inc, 1789 Addison Way, Hayward, CA 94544, USA) and the MA200 (AethLabs, 1640 Valencia St, Suite 2C, San Francisco, CA 94110, USA) use different filter materials. The STAP relies on a quartz-fiber glass filter, whereas the MA200 is based on a PTFE filter. Since their behavior under fast changes of the relative humidity is not described yet, we investigate both instruments in this study.

### 2.2.1 Single Channel Tricolor Absorption Photometer (STAP)

This photometer detects light intensities behind two quartz-fiber glass filter (Pall Life Sciences, Pallflex membrane filters, type E70-2075W) at three wavelengths (450, 525, and 624 nm). On the first filter, the sample filter, the light attenuates due to deposited particulate matter. The second filter, the reference filter, is located downstream of the sample filter and allows blank filter reference light-intensity measurements.

By default, the particle light-absorption coefficient is determined internally using 60 s averages of the raw intensity measurements for both filter spots. Therefore, in Eq. (5), $I(t)$ is defined as

$$I(t) = \frac{I_{\mathrm{smp},\lambda}}{I_{\mathrm{ref},\lambda}}, \tag{7}$$

where $I_{\mathrm{smp}}$ and $I_{\mathrm{ref}}$ are the intensity of light at the certain wavelength $\lambda$ behind the sample (smp) and blank reference (ref) filter, respectively. Nevertheless, all raw measurements are recorded with a time resolution of 1 Hz, allowing a recalculation of $\sigma_{\mathrm{abs}}$ at this time resolution. The volumetric flow is set to one liter per minute (L min$^{-1}$). According to the manual, at an internal averaging interval of 60 s, the measurement uncertainty is specified to 0.2 Mm$^{-1}$. The spot diameter is $\sim 4.8$ mm, which leads to a sample area of $A_{\mathrm{spot}} \sim 1.75 \times 10^{-5}$ m$^2$.

### 2.2.2 MA200

The second instrument used here, the microAeth$^{®}$ MA200, is a small-sized (13.7 cm $\times$ 8.5 cm $\times$ 3.6 cm; CE10 420 g) absorption photometer measuring the attenuation of light at five wavelengths (375, 470, 528, 625, and 880 nm; 625 nm wavelengths are investigated in this study) due to deposited particulate matter on a PTFE filter band.

Similar to the STAP, the MA200 detects light intensities behind a sample and reference spot. The particulate matter samples on a sample spot with a 3 mm diameter, leading to a sample area of $A_{\mathrm{spot}} \sim 0.71 \times 10^{-5}$ m$^2$. The reference spot of the same area allows for blank filter measurements. $M_{\mathrm{eBC}}$ is determined under the assumption that the change of attenuation is proportional to the deposited eBC mass. The measurements were recorded with a 1 Hz time resolution. With DualSpot$^{®}$ technology, the instrument is able to reduce uncertainties related to loading effects of up to 60 % (Holder et al., 2018) but was not functioning at the time of the experiment.

Holder et al. (2018) reported that the measurements slightly depend on RH and $T$ of the aerosol sample. However, they observed concentrations of up to 7 mg m$^{-3}$, at which the observed dependence on humidity and temperature did not influence the measured values significantly. Furthermore, they used another version of the instrument (MA350), which may react differently to changes in humidity and temperature.

### 2.3 Experimental setup

The experimental setup is designed to examine the instrument filters in different states. Unloaded filters and differently loaded filters with black carbon (BC) and ammonium sulfate were investigated. The extent to which fast changes in the relative humidity of the air passing through the filter affect absorption measurements was investigated for these conditions.

A miniCAST burner (model 5200, Jing Ltd.) was used to generate soot (BC) aerosol particles due to combustion of propane. The produced BC particle stream can be diluted according the needs of the customer. A detailed description of the miniCAST is supplied by Jing (1999). Additionally, a solution of ammonium sulfate ((NH$_4$)$_2$SO$_4$; solution concentration of 0.05 g / 80 mL) was nebulized to an aerosol and dried afterwards. Either the ammonium sulfate or the BC aerosol was fed into a 0.5 m$^3$ stainless-steel mixing chamber. A fan within the chamber ensures a well-mixed aerosol.

The scheme of the experimental setup is described in Fig. 1. First, two particle-free, dry (RH = 0 %) air flows were produced. One of the flows was humidified by passing through two glass tubes containing distilled water at room temperature with an inlet and outlet for compressed particle-free air. A maximum relative humidity of $\sim 96$ % was reached. Both the dry and humidified air flows were mixed together with a Swagelok brass T-shaped flow splitter, and it was ensured that the sum of both mass flows exceeded 1 L min$^{-1}$ (controlled by a mass-flow controller). Different RH values were produced according to the ratios of the dry and humidified air. For this, valves with markings indicating the opening state of the valves were used to reproduce a consistent mixing RH. The RH and $T$ of the airflow sampled with the photometer were detected with a temperature and relative-humidity sensor (model HYT939, B+B Thermo-Technik GmbH, 78166 Donaueschingen, Germany) within an accuracy of $\pm 1.8$ % (between 0 % and 90 % RH) and $\pm 0.2$ °C (between 0 and 60 °C). Furthermore, this sensor has a response time $t_{63}$ of $< 10$ s. Additionally, this setup could be used with or without a silica-bead-based dryer before the photometers to examine the extent to which a dryer dampens the effect of relative-humidity changes on the photometer absorption measurements.

Two main setups were used to investigate the effect of changes in real humidity. In the first, the filters of the devices were unloaded and the instruments collected a particle-free airflow with adjustable relative humidity to examine the pure filter effect. In the second, the filters of the devices are loaded to a certain degree, and afterwards they sample particle-free humidified air, which accounts for the combination of both effects, the pure filter effect and the effect induced by the hygroscopic behavior of the particles.

The loading aerosol was split into two streams; the absorption photometers sampled simultaneously from one of these CE12. The other one was sampled with a mobility par-

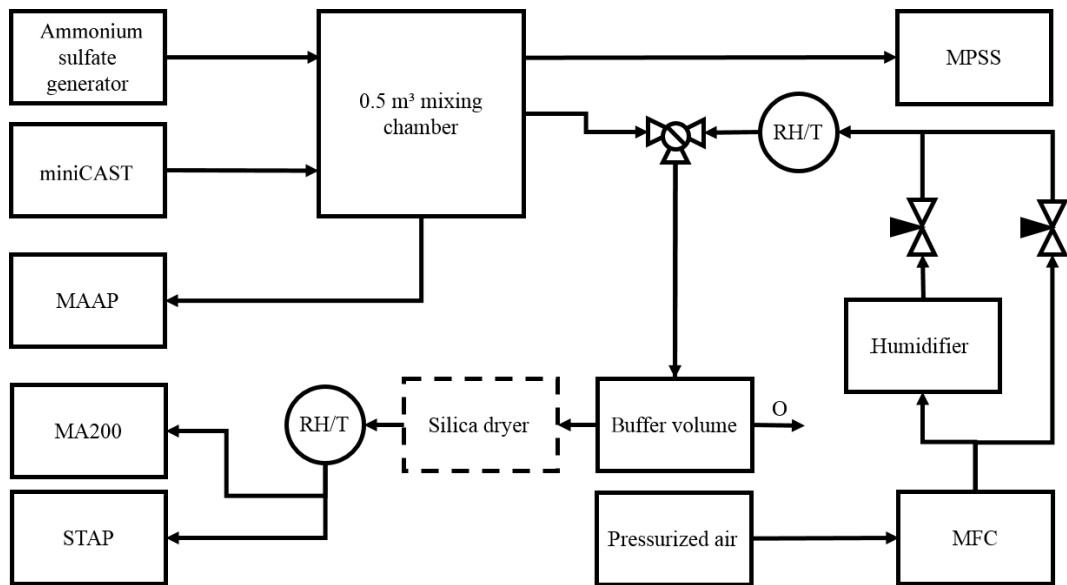

**Figure 1.** CE11 Scheme of the experimental setup. The volumetric flow rates of two air streams were controlled with two needle valves to produce humidified particle-free air via mixing of wet and dry particle-free air. It was ensured that the sum of both flows was larger than the volumetric flow of both absorption photometers investigated here. Any exceeding airflow was directed to a buffer volume with an overflow outlet (O). The relative humidity and the temperature of the air were recorded directly after the humidifier and shortly before the photometers with the RH and $T$ sensors.

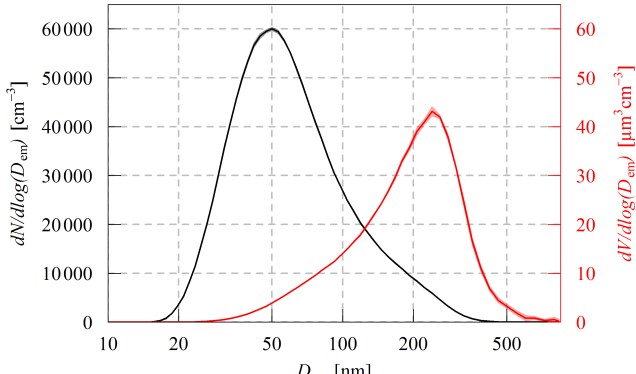

**Figure 2.** Average particle number (black) and volume size distributions (red) of the ammonium sulfate aerosol loaded on the filters in the MA200 and STAP during experiment no. 2 of the ammonium sulfate loading experiments. The average volume of $20.6\,\mu m^3\,cm^{-3}$ is calculated from four scans. The shaded area indicates the standard deviation of the mean. $D_{abs}$ refers herein to the electrical mobility diameter of the aerosol particles.

ticle sizer spectrometer (MPSS; working principle explained in, for example, Wiedensohler et al., 2012) to measure the aerosol-particle number size distribution from which the loading mass was estimated. An example of a generated ammonium sulfate aerosol is shown in Fig. 2. Furthermore, to examine the loading mass concentration of the generated eBC (soot), the generated BC aerosol was also measured with a MAAP.

## 3 Results

This chapter gives an overview of the measurement results. The overall behavior of both instruments is shown for wavelengths of 624 nm in the case of the STAP and 625 nm in the case of the MA200. A closer look at the behavior of both devices at the 1 Hz time resolution shows that both devices differ greatly in quality (see Fig. 3). The STAP (red dots and the smooth fit shown as black line) reacts very fast to relative-humidity changes (dRH / dt as purple line) and then returns relatively fast to the zero line. The MA200, on the other hand, also shows a fast response to relative-humidity changes but then shows a distinct exponential recovery (see Fig. 3; blue dots and smooth fit shown as orange line) and reports absorption coefficients different from zero, although there is no RH change.

Therefore, we use an averaging on a 60 s basis to describe the qualitative behavior of both devices. In the case of the STAP, the internal 60 s averaging is used. For the MA200, on the other hand, a 60 s "running average" is applied to the 1 Hz measurements.

The qualitative behavior of both devices is shown as follows. The corresponding maximum of the excursion of the averaged absorption coefficient ($\Delta\sigma_{abs}$) has been assigned to each absolute change in the relative humidity ($\Delta$RH), where the absolute change of the relative humidity is the difference between the relative humidity at the time of the largest excursion in the absorption coefficient and the relative humidity at

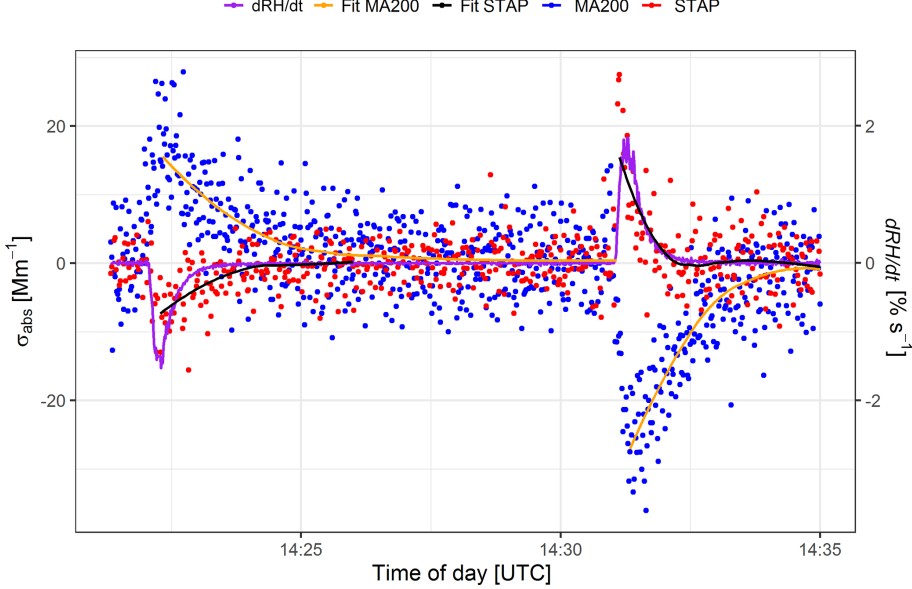

**Figure 3.** One hertz of raw data of $\sigma_{\mathrm{abs}}$ at 625 nm measured by the MA200 (blue points) and recalculated at 624 nm STAP200 (red points), the smooth fit through the measurements (orange and black), and dRH / d$t$ (purple line).

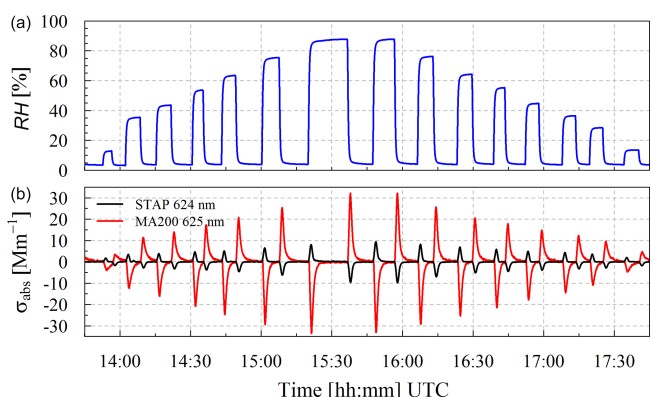

**Figure 4.** Time series of RH **(a)** and absorption coefficient **(b)** measured with STAP (624 nm; black) and MA200 (625 nm; red) with clean filters.

the start of the excursion. This approach also excludes the response time of the RH sensor.

First, the results for the pure filter effect are shown. Afterwards, we present the results of the combined behavior of filter and aerosol particles on the filters. For loaded filters, the combined effect is shown separated into BC and ammonium sulfate loading.

### 3.1   Clean filters

In Fig. 4, the time series of the measured RH upstream of the two photometers (Fig. 4a) and of $\sigma_{\mathrm{abs}}$ measured by the STAP (624 nm) and MA200 (625 nm; in Fig. 4b) are shown. The air sampled by the photometers was entirely particle-free. Rela-

tive humidity was changed in this time series between 3.1 % and 87.7 %. The change rate of RH (dRH / d$t$) was in the range of around $-3.0\,\%\,\mathrm{s}^{-1}$ to $2.9\,\%\,\mathrm{s}^{-1}$. Whereas the measurements of the STAP ranged between $-9.8$ and $9.5\,\mathrm{Mm}^{-1}$, the running 60 s mean of the MA200 readouts ranged from to 32.2 to $-33.6\,\mathrm{Mm}^{-1}$. Furthermore, Fig. 4 shows the opposing behavior of both instruments. Whereas the STAP reacts to positive change rates of the RH with positive particle light-absorption coefficients, the MA200 measures negative particle light-absorption coefficient and vice versa. In summary, this indicates that the different filter materials react oppositely to each other.

Subramanian et al. (2008 TS4) observed that organic matter produced during low-temperature biomass burning has a liquid, bead-shaped appearance when collected on a fibrous filter. Also, these organics can appear as translucent coatings on the filter fibers and therefore change significantly the interaction with incident light. Accordingly, for this study this means that the water in the collection stream can wrap itself around the filter fibers, analogous to the organic materials. Lack et al. (2008) has estimated the bias on filter-based absorption measurement due to loading with organic material. Under conditions with low mass concentrations of organic matter, the agreement with photoacoustic-based aerosol light-absorption measurements was 12 %. Whereas under conditions where the mass concentration of organic material was 15 to 20 times larger than that of light-absorbing carbon, the difference was 50 %–80 %. Therefore, the effect of coating with liquid matter around the fibers is not negligible. In the case of the STAP, the water beads and coating can lead to a higher net reflectance of the filter, which

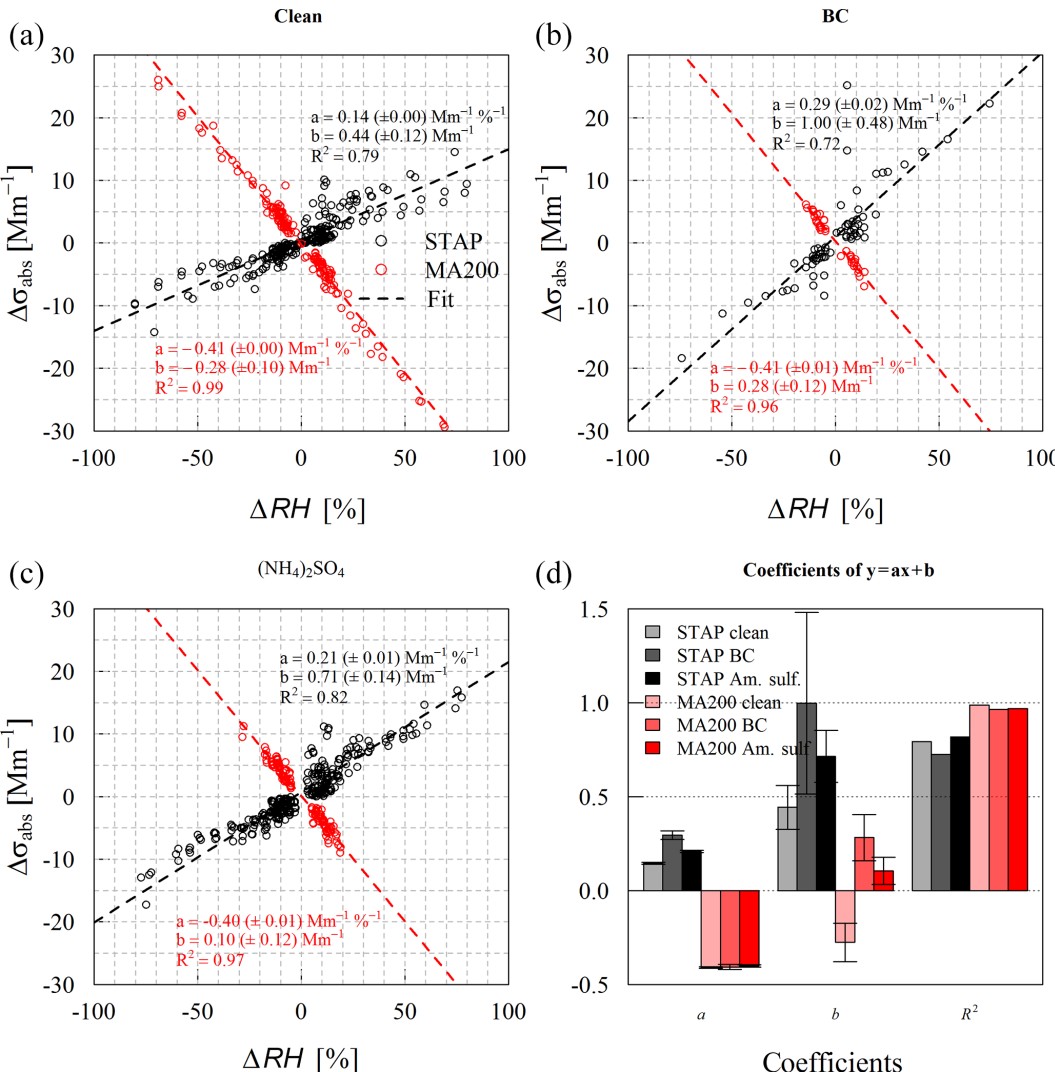

**Figure 5.** Scatterplot (dots) of all observations of the absolute excursion of $\sigma_{abs}$ ($\Delta\sigma_{abs}$) in dependence of the absolute change in RH ($\Delta$RH), its linear regression fit, and the summarizing boxplot of the linear regression fit are shown for the three investigated states (clean, loaded with BC, and ammonium sulfate) at 624 nm (STAP; black colors) and 625 nm (MA200; red colors). Descriptive coefficients are given in Table A1.

appears darker for the photodiode behind the filter. The instrument interprets this as increased attenuation and hence as increased absorption. In addition, the backing material consists of hydrophilic cellulose, which may absorb water under increased relative humidity and thus change its optical properties (Ogren et al., 2017). Compared to the fibrous structure of the quartz-fiber filter, the PTFE filter of the MA200 is a porous, hydrophobic filter. We speculate that these properties result more into a collection of a thin film of water, which could act as an index match of the refractive indices of the PTFE and air. An additional film with an intermediate refractive index reduces the reflectance and thus increases the transmittance, leading to a decreased attenuation. Hence, the instrument interprets decreases in the attenuation as negative absorption.

Since the filter in the STAP reveals a positive and the filter in the MA200 a negative correlation to relative-humidity changes, a combination of both filters within one instrument could account for the observed effect. A new developed instrument could use these two different filter materials on two sampling spots to cancel out the effect of each other. However, more investigations have to be conducted, especially to understand the different recovery behaviors and effect magnitudes of the PTFE and quartz-fiber filter.

The overall behavior of both instruments in the case of clean filters is shown in Fig. 5a. For all investigated $\Delta$RH values, the response behavior of the MA200 ($R^2 = 0.99$) is more stable than the STAP ($R^2 = 0.78$). However, the response is stronger than the average response of the STAP at the presented wavelength. Whereas the STAP shows a dependency of $0.14\,\mathrm{Mm}^{-1}\,\%^{-1}$, which means an increase in

**Table 1.** Filter loading mass concentration ($M_{eBC}$) of the black-carbon particles and filter areal loading density (deposited mass per spot area) $\rho_i^*$. $M_{eBC}$ values were determined by dividing the average $\sigma_{abs}$ of the STAP with an assumed MAC of $6.6\,m^2\,g^{-1}$ or based on the MAAP measurements. Usage of same filter is indicated by separation with thick horizontal lines. Bold written entries were used for the investigation of the RH effect.

| Filter number | $M_{eBC}$ ($\mu g\,m^{-3}$) | $\rho_{eBC,i}^*$ ($mg\,m^{-2}$) | |
| --- | --- | --- | --- |
| | | STAP | MA200 |
| No. 1 | **44.5 (STAP)** | **14.0** | **5.4** |
| | 43.4 (STAP) | 37.9 | 14.4 |
| | **27.6 (STAP)** | **42.9** | **16.3** |
| No. 2 | **52.6 (MAAP, two scans)** | **2.8** | **1.1** |
| No. 3 | – | 13.7 (integral of STAP) | No data |

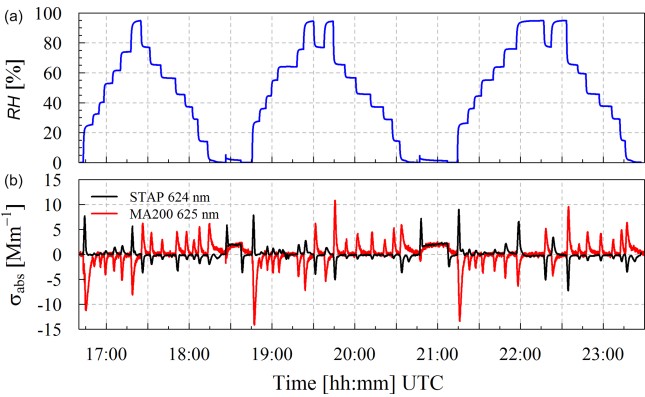

**Figure 6.** RH of the air stream sampled by the MA200 and the STAP **(a)** and $\sigma_{abs}$ measured by MA200 and STAP at 625 nm (624 nm) **(b)**. First up and down ramp of RH conducted with clean filter, and second and third conducted under conditions with filter loaded with ammonium sulfate. Loading periods around 18:30 and 21:00 UTC.

absorption with increasing RH, the MA200 shows an opposing behavior, with a larger absolute value in the slope of $-0.41\,Mm^{-1}\,\%^{-1}$ (Table A1). As shown in Fig. 4, for each device the magnitude of the deviation due to positive or negative changes in humidity is approximately the same.

## 3.2 Loaded filters

Different aerosol types deposit on the filter within the instruments while measuring $\sigma_{abs}$. These aerosol types are either hydrophilic or hydrophobic and hence experience water uptake or not under conditions of elevated RH. Thus, the more particle material is deposited on the filter, the more water deposits on the filter. Therefore, this section shows the influence of different filter loading materials on the RH effect and also points out the effect of different filter loading mass. The observed effect includes both the effect for clean filters (the pure filter effect) and the effect of the hygroscopic behavior of the particles loaded onto the filter. But investigating the effect of the loading material alone is simply not possible, since no filter material exists which is not affected by RH changes. Therefore, the presented results are always a combination of the filter effect and the material effect. For both considered loading materials, the mass loaded onto the filters was calculated by multiplying the prevalent loading mass concentration within the mixing chamber with the volume flow rate of the instrument and the loading duration. The filters were loaded to a certain extent with different materials, and afterwards the absorption photometer was sampling particle-free air with adjustable humidity. The different sample spot areas of the absorption photometers is considered herein by normalizing the loaded mass with the respective sample spot area. This is referred to throughout as filter loading areal density $\rho^*$.

### 3.2.1 Black carbon

During the experiment the eBC loading mass concentration was estimated with different methods depending on the stability of the mass concentration and loading duration and ranged between 27.6 and $52.6\,\mu g\,m^{-3}$. In Table 1, the $\rho^*$ of eBC per spot area ($\rho_{eBC}^*$) of both instruments is shown. During experiment no. 1 (see Table 1) the mean absorption coefficient of the STAP was divided by a MAC of $6.6\,m^2\,g^{-1}$, since the absorption was stable during the loading period, and it is a direct measure from the sampling instrument. For the experiment no. 2 the loading mass concentration was taken from the average of two consecutive MAAP measurements, since the loading period was shorter than 2 min, which is shorter than the internal averaging period of the STAP so that no stable absorption coefficient readouts could be provided by the STAP. During experiment no. 3 no MAAP was available and the absorption coefficient measured by the STAP was unstable. We therefore decided to estimate the loaded eBC mass by integrating the absorption coefficient during the loading period and dividing it by the MAC. Four different $\rho_{eBC}^*$ values were considered in the case of the STAP, and three were considered for the MA200. Due to the smaller volume flow rate of the MA200, $\rho_{eBC}^*$ is in each corresponding case smaller than $\rho_{eBC}^*$ for the STAP, and therefore, if any effects of different loadings are observed, these might not be as distinct as for the STAP.

For all considered BC loading cases the averaged response of STAP and MA200 to relative-humidity changes is shown in Fig. 5b, and corresponding linear fitting and correlation parameters are given in Table A1. The STAP shows a dependency of $0.29\,Mm^{-1}\,\%^{-1}$ in this case. This means the absolute changes in RH affecting the BC loaded filter lead to stronger $\sigma_{abs}$ deviations than in the clean case. For the

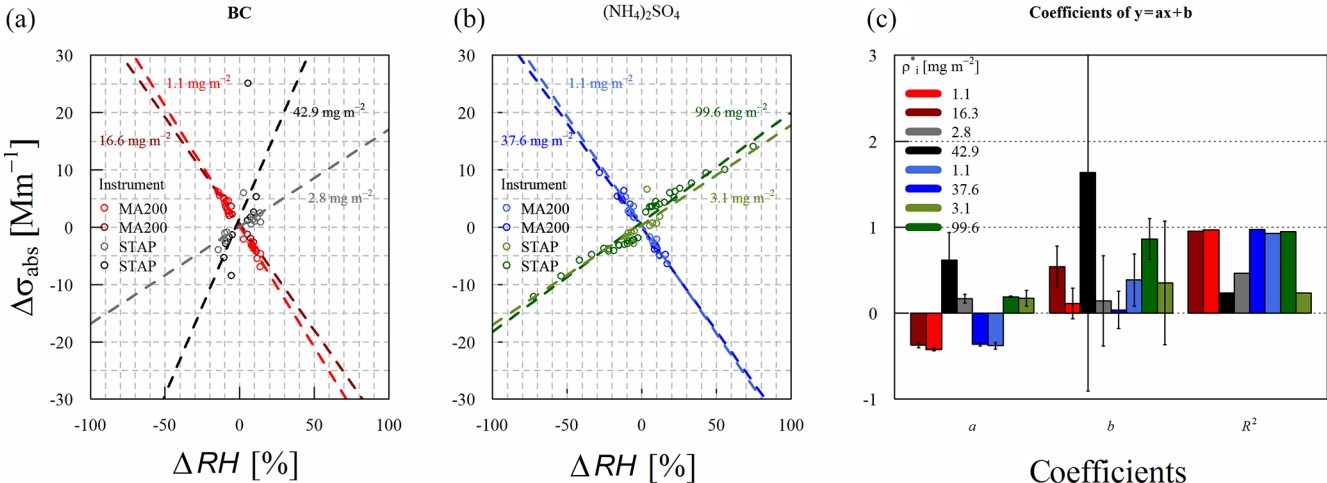

**Figure 7.** Scatterplot of change in absorption ($\Delta\sigma_{abs}$) in dependence of the absolute change in RH ($\Delta$RH) separated into the different loading states (loaded with BC and ammonium sulfate) and minimum and maximum loading areal density on the filter. Dashed and colored lines represent the linear regression fit. Red and blue colors indicate MA200 at 625 nm, and black and green colors indicate STAP at 624 nm. In **(a)** BC loading is shown, whereas in **(b)** the ammonium sulfate case is displayed. Coefficients of the linear regression fit are displayed in **(c)**. Colored shading in the linear fits and of the points are same as in **(c)**.

MA200, the response of the instrument to rapid changes in RH does not depend on the loading material for the filter with BC, since the regression slope is the same as in the clean case. However, the $y$ intersects deviate from each other but within $0.6\,\mathrm{Mm^{-1}}$. We assume that this could be due to the lower absolute loading on the filter (15 % of the STAP loading because of $0.15\,\mathrm{L\,min^{-1}}$ flow rate compared to $1\,\mathrm{L\,min^{-1}}$ of the STAP) or to the MA200 response being in general independent of the filter loading material.

Considering different loading areal densities, the MA200 shows more or less the same behavior (see Table A1 and Fig. 7). The slope of the linear correlation fit ranges from $-0.42$ to $-0.37\,\mathrm{Mm^{-1}\,\%^{-1}}$ for corresponding loading areal densities of 1.1 to $16.3\,\mathrm{mg\,m^{-2}}$. The STAP shows a larger variability (Fig. 7a; black and gray colors). For a $\rho^*$ of 2.8 to $42.9\,\mathrm{mg\,m^{-2}}$, the STAP response ranges from 0.17 to $0.62\,\mathrm{Mm^{-1}\,\%^{-1}}$. However, these results may not be entirely representative, since $R^2$ is only larger than 0.9 (0.94) for a loading density of $13.7\,\mathrm{mg\,m^{-2}}$. Maybe the smaller amount of data points in the other cases explains this. However, the number of data points observed with the MA200 was also small, but $R^2$ is in each case larger than 0.9. Similar to the clean case, for both instruments, drying and humidifying the sample stream to the same extent resulted in a deviation of $\sigma_{abs}$ with the similar magnitude.

### 3.2.2 Ammonium sulfate

The loading mass concentration of the $(NH_4)_2SO_4$ aerosol depositing on the filters was estimated by integrating the mean ammonium sulfate aerosol-particle volume size distribution of the loading period and multiplying this volume con-

**Table 2.** Average volume and mass concentration ($V_{(NH_4)_2SO_4}$, $M_{(NH_4)_2SO_4}$) of the loading $(NH_4)_2SO_4$ aerosol derived from the used MPSS (number of used scans in brackets) and loading areal density $\rho^*_{(NH_4)_2SO_4}$ of the filters are given. Usage of same filter is indicated by a separation with thick horizontal lines, which means that the filter loading mass was adding up during the experiments.

| Filter number | $V_{(NH_4)_2SO_4}$ ($\mu m^3\,cm^{-3}$) (no. of scans) | $M_{(NH_4)_2SO_4}$ ($\mu g\,m^{-3}$) | $\rho^*_{(NH_4)_2SO_4}$ (mg m$^{-2}$) STAP | MA200 |
|---|---|---|---|---|
| No. 1 | 15.4 (2) | 27.2 | 3.1 | 1.2 |
|  | 18.6 (1) | 32.9 | 10.5 | 4.0 |
|  | 20.6 (3) | 36.4 | 31.3 | 11.9 |
| No. 2 | 20.6 (4) | 36.5 | 40.8 | 15.5 |
| No. 3 | 33.1 (3) | 58.6 | 32.5 | 12.4 |
|  | 33.5 (5) | 59.3 | 98.7 | 37.6 |
| No. 4 | 20.3 (3) | 36.0 | 21.1 | 8.0 |
|  | 20.3 (3) | 36.0 | 41.9 | 15.9 |
| No. 5 | 23.9 (3) | 42.4 | 28.9 | No data |
|  | 28.4 (4) | 50.2 | 69.8 | No data |
|  | 29.8 (2) | 52.8 | 99.6 | No data |

centration (see Table 2) with an assumed ammonium sulfate density of $1.77\,\mathrm{g\,cm^{-3}}$ (Haynes, 2014). The loading mass concentrations were in the range of 27.2 to $59.3\,\mathrm{\mu g\,m^{-3}}$ (see Table 2). The very narrow standard deviation around the mean particle number and volume size distribution in Fig. 2 indicate clearly that the loading mass concentrations were very stable during the loading periods.

In the experiments, ammonium sulfate filter loading areal densities were 3.1 to 99.6 mg m$^{-2}$ in the case of the STAP and 1.2 to 15.9 mg m$^{-2}$ in the case of the MA200. Exemplarily, particle number size distribution and particle volume size distribution of the ammonium sulfate aerosol of the loading on 23 February are shown in Fig. 2. It is clearly visibly that the ammonium maximum in the particle volume size distribution peaked around a mobility diameter of 260 nm.

Figure 6 shows exemplarily the time series of the sample air RH and of the $\sigma_{abs}$ measured with STAP and MA200 operated with clean filters. A RH of 0.0 % to 96.2 % with humidity change rates of $-1.42 \% \, s^{-1}$ to $1.09 \% \, s^{-1}$ was measured. Compared to the case in Fig. 4, here a stepwise change of RH is shown. These steps resulted in a smaller absolute excursion of $\sigma_{abs}$ which ranges from $-7.2$ to $9.0$ Mm$^{-1}$ (STAP; 624 nm, 60 s measurement resolution) and $-14.1$ to $10.9$ Mm$^{-1}$ (MA200; 625 nm, 60 s running mean). Furthermore, Fig. 6 shows the response of the $\sigma_{abs}$ to RH changes at three different states of filter loading. During the first ramp, the filters were clean, during the second period the filters had a filter areal loading density of 32.5 mg m$^{-2}$ (STAP) and 12.4 mg m$^{-2}$ (MA200), and during the third ramp the filter in the STAP had a loading areal density CE13 of 98.7 mg m$^{-2}$, and the MA200 filter was loaded with an areal loading density of 37.6 mg m$^{-2}$. The response of the instruments during these periods is shown in Table A1.

In Fig. 5c, the overall (mean) response of both instruments to RH changes is shown in the case of loading with ammonium sulfate. The MA200 behaves similarly to the clean and BC case (slope of $-0.40$ Mm$^{-1}$ %$^{-1}$). The $\sigma_{abs}$ measured by the STAP responses opposingly, with a positive slope of $0.21 \pm 0.01$ Mm$^{-1}$ %$^{-1}$, which is roughly half of the amplitude shown by the MA200 and around two-thirds of the BC loaded case.

As shown in Fig. 6, both absorption photometers measure an "apparent" absorption coefficient of approximately 2 Mm$^{-1}$ during loading with ammonium sulfate (18:30 and 21:00 UTC). This shows that absorption photometers react sensitively to scattering aerosols such as ammonium sulfate. The scattering ability of any material can be described with the real part of its refractive index. It seems that for the STAP the slope of the correlation increases with increasing scattering of the loading material (0.15 Mm$^{-1}$ %$^{-1}$ for a clean filter, 0.21 Mm$^{-1}$ %$^{-1}$ for ammonium sulfate, and 0.30 Mm$^{-1}$ %$^{-1}$ for BC). Ammonium sulfate has a real part of $1.521 \pm 0.002$ (at 532 nm; Dinar et al., 2007), and BC from combustion processes has a real value CE14 of 1.96 at 530 nm (Kim et al., 2015, following Ackermann and Toon, 1981). Hence, the quartz-fiber glass filters loaded with "artificially" absorbing aerosol inside the STAP could lead to a variation in the response to relative-humidity changes. But the MA200 was loaded with ammonium sulfate as well, and its response to relative-humidity changes is almost constant for all considered loading materials. Therefore the observation is caused by the interaction of quartz-fiber glass filters with

the loading material and the PTFE filters inside the MA200 do not cause this behavior, the filter loading of the MA200 was too low or there are other mechanisms explaining this. Furthermore, since only three different cases (clean, ammonium sulfate, and BC) were observed in this study, more materials should be considered to investigate this phenomenon. No correlation of linear regression slope and filter loading areal density $\rho^*$ was observed in either case, for the STAP and the MA200, respectively. The slope ($a$) of STAP ranges from 0.17 to 0.24 Mm$^{-1}$ %$^{-1}$, and the slope of the MA200 ranges from $-0.36$ to $-0.42$ Mm$^{-1}$ %$^{-1}$. With a relative difference from the minimum to maximum slope of 15.2 %, the response of the MA200 is less variable than that of the STAP, with a relative variability of 28.6 % (slopes with $R^2 < 0.8$ excluded, all points included 36.8 %). In Fig. 7b, the spread of the slopes within the shown cases is exemplarily shown for the investigated minimum and maximum load of the filters. Overall, the magnitude of the deviation of $\sigma_{abs}$ was independent of the sign of humidity change for both instruments.

### 3.3 Correction approach

The above chapters describe the overall behavior of the instruments to relative-humidity-change averaging time of 60 s. To correct for the described effect, a 1 Hz time resolution is needed to resolve the instantaneous response of the instruments to relative-humidity changes. For this purpose, a further laboratory experiment was conducted in which the inlets of both instruments could be flexibly exposed to humidified air. In our particular case, we hold the inlet in a beaker with a moistened tissue. In order to avoid any dampening bias, all measurements were conducted without a particle filter in front of the inlet. But during this experiment a background of about $\sigma_{abs}$ of 1.2 Mm$^{-1}$ was measured so that the filter loading was very low. First, we consider the STAP, and afterwards the MA200 is investigated.

### 3.3.1 STAP

In Fig. 8 the correlation of the RH-change rate ($\mathrm{dRH} / \mathrm{d}t$) and the measured $\sigma_{abs}$ at 624 nm measured by the STAP (red circles) and recalculated with respect to standard conditions (pressure of 1013.25 hPa and temperature 273.15 K) is shown. The STAP-based background eBC mass concentration during the experiment was $\sim 190$ ng m$^3$ (at standard conditions, $\sigma_{abs}$ at 624 nm converted with a MAC of 6.6 m$^2$ g$^{-1}$), which corresponds to offset (standard conditions corrected values) in the shown scatterplot of Fig. 8 and which has no influence on the response to RH changes as shown previously.

The RH-change rate ranged from $-10.8 \% \, s^{-1}$ to $14.5 \% \, s^{-1}$. These rates correspond to a $\sigma_{abs}$ of $-231$ to $192$ Mm$^{-1}$ for recalculated values at standard conditions and $-203$ to $164$ Mm$^{-1}$ directly measured by the instrument. But these measurements are biased by the response time of the relative-humidity sensor so that the "real" RH-change rate

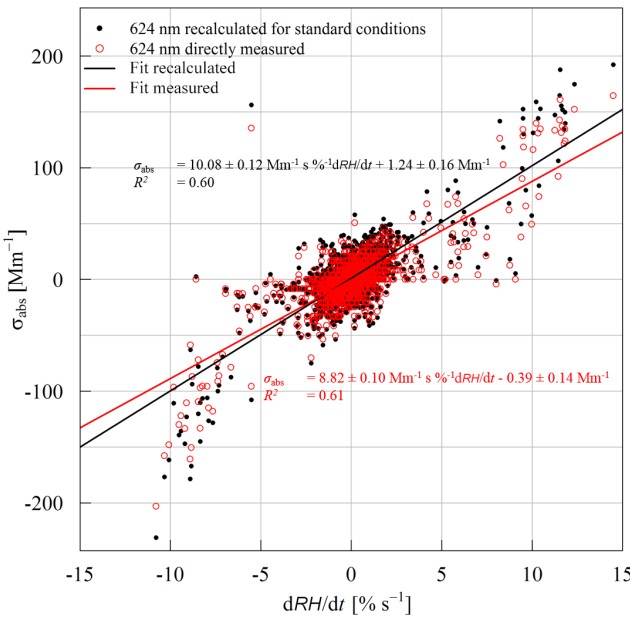

**Figure 8.** Scatterplot of $\sigma_{\mathrm{abs}}$ and change rate of RH (dRH / d$t$) at 624 nm derived directly from STAP without any corrections (red) and recalculated $\sigma_{\mathrm{abs}}$ at 624 nm, including corrections to standard conditions (black). Linear fit equations and correlation coefficient are given in the corresponding colors.

cannot be fully represented by these measurements. On average the slope (correction factor $C_{\mathrm{RH}}$ in Eq. 8) of the linear fit is $10.08\,(\pm 0.12)\,\mathrm{Mm^{-1}\,s\,\%^{-1}}$ for standard conditions and $8.82\,(\pm 0.10)\,\mathrm{Mm^{-1}\,s\,\%^{-1}}$ for direct instrument output. Calculating the particle light-absorption coefficient introduced by RH changes with

$$\sigma_{\mathrm{abs,RH}} = C_{\mathrm{RH}}\frac{\mathrm{dRH}}{\mathrm{d}t}, \tag{8}$$

and for different RH-change rates in both the recalculated and direct instrument correcting for the observed effect as follows:

$$\sigma_{\mathrm{abs,corr}} = \sigma_{\mathrm{abs,meas}} - \sigma_{\mathrm{abs,RH}}, \tag{9}$$

and after replacing $\sigma_{\mathrm{abs,RH}}$ in Eq. (8) with Eq. (9),

$$\sigma_{\mathrm{abs,corr}} = \sigma_{\mathrm{abs,meas}} - C_{\mathrm{RH}}\frac{\mathrm{dRH}}{\mathrm{d}t}. \tag{10}$$

The $y$ intersect of the linear fit in Fig. 8 must not be considered for correction, as mentioned before. Disadvantageously, with this correction the noise of the RH sensor will propagate in the corrected $\sigma_{\mathrm{abs}}$. Furthermore, the linear fit in Fig. 8 under- or overestimates the behavior in regimes of very high relative-humidity-change rates, most likely due to the response time of the RH sensor, so that the correction function cannot entirely correct the bias. Therefore, the given correction factor $C_{\mathrm{RH}}$ consists of uncertainties which cannot be

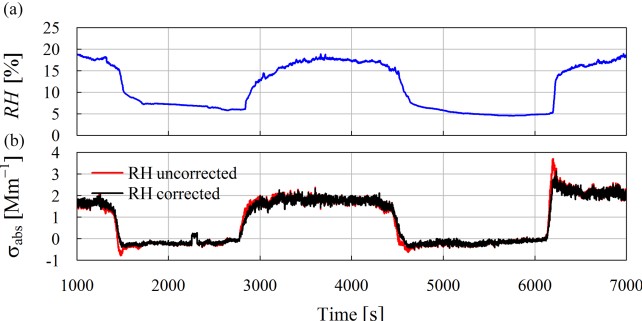

**Figure 9.** Time series of RH before the inlet of the STAP (blue; **a**) and recalculated $\sigma_{\mathrm{abs}}$ (running 60 s mean of 1 Hz calculations) at standard conditions corrected (black) and uncorrected (red) for RH changes (**b**).

entirely addressed. Hence, it is only a first guess and needs further refinement, and now we do not recommend using the correction approach as long the uncertainties are not fully addressed. Furthermore, since only one STAP was tested, other STAPs may have other correction factors due to a unit-to-unit variability. Additionally, other filter materials used in the STAP can also lead to another behavior. In any case, the upper function was applied to STAP measurements conducted with the same RH sensor under atmospheric conditions.

Exemplarily, Fig. 9 shows this application. The figure shows airborne measurements of $\sigma_{\mathrm{abs}}$ at 624 nm derived with the STAP derived during a campaign conducted in March 2017 in eastern Germany. Figure 9a displays the RH of a dried aerosol sample stream measured upstream of the STAP. Figure 9b shows the recalculated $\sigma_{\mathrm{abs}}$ at the 624 nm wavelength corrected for RH changes (black) and biased by RH changes (red). In the periods where the RH changes relatively fast (dRH / d$t$ of $-0.55\,\%\,\mathrm{s^{-1}}$ to $0.56\,\%\,\mathrm{s^{-1}}$, for example, at around 6200 s), the uncorrected $\sigma_{\mathrm{abs}}$ overshoots. The correction significantly reduces this bias and smooths out the measurements during the periods of RH changes. At the peaks of dRH / d$t$, the difference of the corrected and uncorrected values is up to $1.5\,\mathrm{Mm^{-1}}$, which is significant with respect to the measured $\sigma_{\mathrm{abs}}$. The periods with negative $\sigma_{\mathrm{abs}}$ are not introduced by the RH effect. We moreover think that a small offset is introduced in the initialization process of the instrument. Despite the imperfection of the correction scheme, this linear approach can be useful for deriving a rough estimate of the accuracy of the measurements. For instance, if we let $x$ be the required accuracy for the measurements (in %) and $\sigma_{\mathrm{abs}}$ the measured particle light-absorption coefficient, we can express the minimum ambient particle light-absorption coefficient which is needed to fulfill the accuracy criterion in dependency of CE15 the RH-change rate dRH / d$t$:

$$\sigma_{\mathrm{abs,meas}} \geq \frac{100\,\%}{x\,(\%)}C_{\mathrm{RH}}(\mathrm{Mm^{-1}\,s\,\%^{-1}})\left|\frac{\mathrm{dRH}}{\mathrm{d}t}\right|(\%\,\mathrm{s^{-1}}). \tag{11}$$

Exemplarily, if a change rate of $0.1\,\%\,\mathrm{s}^{-1}$ is measured and an accuracy of $25\,\%$ is needed, a measured particle light-absorption coefficient of at least around $4\,\mathrm{Mm}^{-1}$ is needed to fulfill the accuracy criterion.

### 3.3.2 MA200

The exponential recovery behavior of the MA200 (see Fig. 3) requires a more complex approach to correct for relative-humidity changes. Therefore, the apparent particle light-absorption coefficient can be described as a function of $\mathrm{dRH}/\mathrm{d}t$ at a given time $t\,(\sigma_{\mathrm{abs,RH}}(t))$:

$$\sigma_{\mathrm{abs,RH}}(t) = a\frac{\mathrm{dRH}}{\mathrm{d}t}(t) + b\sigma_{\mathrm{abs,RH}}(t-1), \tag{12}$$

where $a$ is a linear factor describing the dependency of $\sigma_{\mathrm{abs,RH}}$ on $\mathrm{dRH}/\mathrm{d}t$ and $b$ is an exponential decay parameter between 0 and 1. The Eq. (12) corresponds to an autoregressive moving average model with an exogenous variable (ARMA-X).

The function *marima* of the *R* package *marima* (v2.2) is capable of deriving such an ARMA-X model (details in Appendix A). From this, the coefficients $a$ and $b$ can be derived. These can be furthermore used as initial parameters for an optimization by minimizing the sum of the squared residual errors. The derived ARMA-X model describes $\sigma_{\mathrm{abs}}(\mathrm{dRH}/\mathrm{d}t)$ as follows:

$$\sigma_{\mathrm{abs,RH}}(t) = -0.47(\mathrm{Mm}^{-1}\,\mathrm{s}\,\%^{-1})\frac{\mathrm{dRH}}{\mathrm{d}t}(t) + 0.93\sigma_{\mathrm{abs,RH}}(t-1), \tag{13}$$

and with the applied optimization,

$$\sigma_{\mathrm{abs,RH}}(t) = -0.50(\mathrm{Mm}^{-1}\,\mathrm{s}\,\%^{-1})\frac{\mathrm{dRH}}{\mathrm{d}t}(t) + 0.96\sigma_{\mathrm{abs,RH}}(t-1). \tag{14}$$

Figure 10 shows time series of RH (Fig. 10a) and 60 s running average $\sigma_{\mathrm{abs}}$ derived with the MA200 at 625 nm with the 1 Hz time resolution derived during the laboratory experiment mentioned in Sect. 3.4. Under the influence of $\mathrm{dRH}/\mathrm{d}t$ in the range of $-11.2\,\%\,\mathrm{s}^{-1}$ to $17.1\,\%\,\mathrm{s}^{-1}$, the 60 s running average of $\sigma_{\mathrm{abs}}$ is between $-6.5$ and $7.7\,\mathrm{Mm}^{-1}$ ($M_{\mathrm{eBC}}$ equivalent of $-0.99$ to $1.20\,\mu\mathrm{g}\,\mathrm{m}^{-3}$). Subtracting the calculated apparent particle light-absorption coefficient in dependency of RH changes following Eqs. (13) and (14) to this dataset, $\sigma_{\mathrm{abs}}$ shrinks to $-3.2$ from $4.7\,\mathrm{Mm}^{-1}$ or $-1.0$ from $3.7\,\mathrm{Mm}^{-1}$ in the non-optimized and optimized case, respectively. This corresponds to an $M_{\mathrm{eBC}}$ of around $-0.5$ to $0.7\,\mu\mathrm{g}\,\mathrm{m}^{-3}$ or $-0.2$ to $0.6\,\mu\mathrm{g}\,\mathrm{m}^{-3}$ in the optimized case. This indicates that the presented approach can significantly reduce the RH bias in the presented case. But RH-change-induced fluctuations in $\sigma_{\mathrm{abs}}$ are still visible, which indicates that the correction scheme cannot account entirely for all the bias introduced by a change in RH. Here, the response time of the sensor could

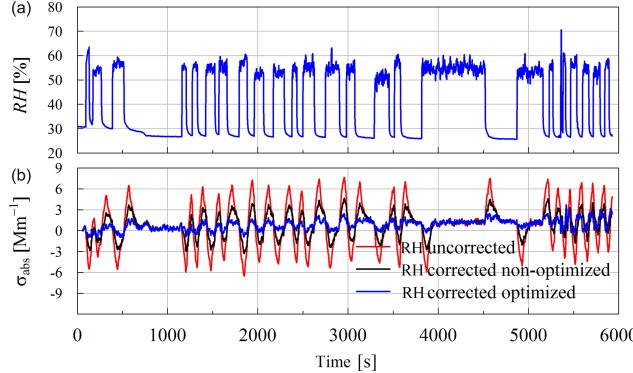

**Figure 10.** Time series of a laboratory measurement of $\sigma_{\mathrm{abs}}$ conducted with the MA200 without a filter. **(a)** shows the RH of the aerosol sample, and **(b)** displays the 60 s running average of measured $\sigma_{\mathrm{abs}}$ at 1 Hz time resolution and 625 nm uncorrected and biased by RH changes (red) and corrected with the modeled $\sigma_{\mathrm{abs}}$ derived with the ARMA-X model (black).

account at least for a part of the imperfection of the correction approach and cannot be fully quantified yet.

Unfortunately, the application of the same correction approach to other similar experiments resulted in different correction function $a$ and $b$. Applying the approach to two clean case experiments from Sect. 3.1 resulted in optimized parameters of $a = -0.92$ and $-1.03$ and $b = 0.974$ and $0.971$. Hence, it is just a first step in trying to account for relative-humidity changes, and further research with more MA200 photometers simultaneously has to be done to fully understand the underlying processes and to fully quantify the uncertainties of the correction scheme. Nevertheless, the presented approach significantly reduces the amplitude of the bias in the shown dataset (see Fig. 10). However up until now we cannot recommend using the given parameters to correct for RH effects. At most it can be used to make a rough estimate of how measurements of the particle light-absorption coefficient derived from the MA200 could be biased by RH changes.

## 4 Summary and conclusion

Here we presented a unique set of laboratory studies to investigate the response of two different types of filter-based absorption photometers (STAP and MA200) with different filter materials (quartz-fiber glass and PTFE) to relatively fast changes in relative humidity of sampled aerosol. Different filter loading densities with different loading material (clean, black carbon, and ammonium sulfate) were considered in this study. Both instruments revealed that they react to fast humidity changes but in opposite ways, induced by the different filter material. This opposing behavior could be a chance to design an instrument on the basis of both filter materials so that the effects cancel out each other. No significant differ-

ences between the loading aerosol types were observed in the case of the MA200, whereas the STAP revealed the strongest response in the BC case.

The MA200 revealed a very robust response of $-0.42$ to $-0.36\,\mathrm{Mm^{-1}\,\%^{-1}}$ (negative excursion of $\sigma_{\mathrm{abs}}$ with increasing RH), whereas the STAP fluctuated more across loading areal density and loading aerosol type, with a positive excursion of $\sigma_{\mathrm{abs}}$ in the range of 0.17 to $0.62\,\mathrm{Mm^{-1}\,\%^{-1}}$. We assume that in the case of the MA200, it is more of a filter effect or that the filter loadings were too low to have a significant effect due to the lower volume flow rate. For the STAP, more parameters could also have an effect, and further investigation is needed. For loading areal density on the filter, no correlation was found, although we expected that the hygroscopic ammonium sulfate would affect the transmissivity of the filter-aerosol layer. Hence, we think that excursions of $\sigma_{\mathrm{abs}}$ due to relative-humidity changes are mainly caused by water vapor filter-material interactions and are independent of the filter loading areal density or they were too low to observe significant effects.

Furthermore, we developed some correction approaches for both instruments to account for fast RH changes. For the STAP a linear correction function was derived. This correction follows a linear approach, including a correction factor of $C_{\mathrm{RH}} = 10.08 \pm 0.12\,\mathrm{Mm^{-1}\,s\,\%^{-1}}$ for standard conditions and $8.82 \pm 0.10\,\mathrm{Mm^{-1}\,s\,\%^{-1}}$ for direct instrument output without any corrections. But this correction was created with a RH sensor with a response time which introduces a bias in the correction approach and cannot be quantified yet. Also, other RH sensors might result in a different correction formula. Exemplarily, this correction was applied to an airborne dataset and has shown promising results. For the MA200 no linear correction function can be provided, since after an excursion of $\sigma_{\mathrm{abs}}$ the MA200 shows a distinct exponentially behaving recovery function. Therefore, an ARMA-X model was developed to account for this exponential decay and to describe the $\sigma_{\mathrm{abs}}$ in dependency of dRH / d$t$. Applying this to the presented dataset, this significantly reduced the excursion introduced by RH changes. We do not recommend using the presented approaches, since the uncertainties cannot be fully quantified yet and a refinement is needed on the basis of more experiments to fully understand the underlying processes and to quantify the uncertainties. The findings summarized above lead to the following recommendations on how to use this type of instrument.

1. When used for vertical profiling, apparent sharp gradients in RH during the profile have to be considered.

    (a) The ascending speed of the profiling platform should be reduced, if possible, to decrease the temporal change of RH, but in some scenarios this is simply not possible.

    (b) Therefore, when fast relative-humidity changes cannot be avoided, such periods have to be removed

from the dataset or the uncertainties of the measurements based on the presented correction functions must at least be estimated.

2. We thus recommend recording the RH of the sampled aerosol. This allows us to determine RH-change rates and to roughly estimate the bias of RH changes on filter-based absorption measurements with these two instruments.

3. The usage of a dryer is highly recommended because it reduces the amplitude of the excursion in the measurements during fast RH changes.

4. For both instruments we recommend conducting more similar experiments to address the flaws of our study in order to refine the presented correction approaches.

5. Since the response is different in magnitude and sign for both filter materials, we recommend examining the effect for other filter materials as well.

*Code and data availability.* Any used code and the data can be requested via the given corresponding e-mail address. CE16

## Appendix A: Multivariate autoregressive integrated moving average model with an exogenous variable – MARIMA-X

The multivariate autoregressive integrated moving average model with an exogenous variable (MARIMA-X$(p,d,q)$) can model the behavior of an observation driven by an exogenous variable. It consists of three parts, namely the autoregressive (AR) part of order p CE17, the moving average (MA) part of order $q$, and the integrating ($I$) part, which describes how often ($d$ times) a time series has to be differentiated to be stationary. An MARIMA-X model can be described as

$$
\begin{aligned}
Y_t = a X_t + b_1 Y_{t-1} + b_2 Y_{t-2} + \ldots + b_n Y_{t-n} \\
+ \epsilon_t + c_2 \epsilon_{t-2} + c_1 \epsilon_{t-1} + \ldots + c_n \epsilon_{t-n},
\end{aligned} \tag{A1}
$$

with $Y_t$ being the predicted value of the model at the time $t$. $b_1 Y_{t-1}$ to $b_n Y_{t-n}$ are part of the autoregressive module of the model, with the corresponding coefficients $b_1$ to $b_n$, describing the contribution of each $Y_{t-n}$ to $Y_t$. $X_t$ represents the corresponding independent exogenous variable at time $t$, whereas the $\epsilon_t$ values are part of the moving average of the model, which accounts for lagged error terms, $\epsilon_t$, introduced by the model itself. $c_1$ to $c_n$ indicate the contribution of $\epsilon_t$ to $\epsilon_{t-n}$ to $Y_t$. For predictions of a variable, the error term is unknown. A special case of the MARIMA-X model is the MARMA-X model or ARMA-X, in which the integrating part has an order of zero. Detailed information about MARIMA models can be found in Durbin and Koopman (2012) and Lütkepohl (2005). A tutorial for estimating MARIMA models in $R$ is provided by Spliid (2016).

## Appendix B: Table with overview of all investigated cases

**Table B1.** Coefficients of the linear regression of the instrument response to RH changes ($a$ is slope, $b$ is $y$ intersect, and $R^2$ is the correlation coefficient) for the clean, ammonium sulfate, and BC case for different loading areal densities $\rho_i^*$. The number of data points is represented by $n$. Bold written entries represent cases with a $R^2$ larger than 0.8. $\sigma$ indicates the standard deviation of the fitting parameters. For ammonium sulfate, $\tau$ not considered.

| Loading aerosol | Device | $\rho_i^*$ (mg m$^{-2}$) | $\tau$ | $n$ | $a$ (Mm$^{-1}$ %$^{-1}$) | $\sigma(a)$ (Mm$^{-1}$ %$^{-1}$) | $b$ (Mm$^{-1}$) | $\sigma(b)$ (Mm$^{-1}$) | $R^2$ |
|---|---|---|---|---|---|---|---|---|---|
| Clean | MA200 (625 nm) | **–** | **–** | **147** | **−0.41** | **0.00** | **−0.28** | **0.10** | **0.99** |
| | STAP (624 nm) | – | – | 241 | 0.14 | 0.00 | 0.44 | 0.12 | 0.79 |
| BC | MA200 (625 nm) | **1.1** | **0.98** | **18** | **−0.42** | **0.02** | **0.11** | **0.18** | **0.97** |
| | | **5.3** | **0.95** | **9** | **−0.40** | **0.03** | **0.38** | **0.24** | **0.96** |
| | | **16.3** | **0.82** | **9** | **−0.37** | **0.03** | **0.54** | **0.24** | **0.95** |
| | MA200 (625 nm, all) | **–** | **–** | **36** | **−0.41** | **0.01** | **0.28** | **0.12** | **0.96** |
| | STAP (624 nm) | 2.8 | 0.93 | 13 | 0.17 | 0.05 | 0.14 | 0.52 | 0.47 |
| | | **13.7** | **0.78** | **33** | **0.29** | **0.01** | **1.26** | **0.37** | **0.94** |
| | | 14.0 | 0.74 | 10 | 0.38 | 0.19 | 0.89 | 1.49 | 0.25 |
| | | 42.9 | 0.52 | 10 | 0.62 | 0.32 | 1.64 | 2.54 | 0.23 |
| | STAP (624 nm, all) | – | – | 66 | 0.29 | 0.02 | 1.00 | 0.48 | 0.72 |
| (NH$_4$)$_2$SO$_4$ | MA200 (625 nm) | **1.1** | **–** | **9** | **−0.38** | **0.04** | **0.39** | **0.31** | **0.93** |
| | | **4.0** | **–** | **10** | **−0.42** | **0.04** | **0.00** | **0.27** | **0.94** |
| | | **8.0** | **–** | **41** | **−0.41** | **0.01** | **0.10** | **0.13** | **0.97** |
| | | **11.9** | **–** | **10** | **−0.40** | **0.03** | **−0.21** | **0.24** | **0.95** |
| | | **12.4** | **–** | **15** | **−0.39** | **0.01** | **0.25** | **0.16** | **0.99** |
| | | **15.5** | **–** | **9** | **−0.37** | **0.03** | **0.47** | **0.23** | **0.95** |
| | | **15.9** | **–** | **18** | **−0.42** | **0.02** | **0.07** | **0.20** | **0.98** |
| | | **37.6** | **–** | **15** | **−0.36** | **0.02** | **0.04** | **0.21** | **0.97** |
| | MA200 (625 nm, all) | **–** | **–** | **127** | **−0.40** | **0.01** | **0.10** | **0.07** | **0.97** |
| | STAP (624 nm) | 3.1 | – | 10 | 0.17 | 0.09 | 0.35 | 0.72 | 0.23 |
| | | 10.5 | – | 10 | 0.19 | 0.09 | 0.33 | 0.72 | 0.28 |
| | | 21.1 | – | 34 | 0.20 | 0.04 | 0.73 | 0.49 | 0.42 |
| | | **28.9** | **–** | **90** | **0.23** | **0.01** | **0.92** | **0.34** | **0.81** |
| | | 31.3 | – | 10 | 0.19 | 0.09 | 0.34 | 0.71 | 0.28 |
| | | **32.5** | **–** | **13** | **0.19** | **0.02** | **0.35** | **0.36** | **0.82** |
| | | 40.8 | – | 10 | 0.21 | 0.10 | 0.42 | 0.79 | 0.28 |
| | | 41.9 | – | 16 | 0.20 | 0.05 | 0.16 | 0.64 | 0.47 |
| | | **69.8** | **–** | **56** | **0.19** | **0.01** | **0.89** | **0.16** | **0.96** |
| | | **98.7** | **–** | **14** | **0.24** | **0.03** | **0.28** | **0.39** | **0.86** |
| | | **99.6** | **–** | **33** | **0.19** | **0.01** | **0.87** | **0.24** | **0.95** |
| | STAP (624 nm, all) | **–** | **–** | **296** | **0.21** | **0.01** | **0.71** | **0.14** | **0.82** |

*Author contributions.* Conceptualization: SD and TM; Data curation: SD; Investigation: SD; Methodology: SD; Software, SD and AS; Supervision, TM, BW, AS, and Alfred Wiedensohler; Visualization: SD; Writing – original draft, SD; Writing – review & editing: SD, TM, BW, and AS.TS5

*Competing interests.* The authors declare that they have no conflict of interest.

*Acknowledgements.* We would like to thank Sascha Pfeiffer very much for his guidance in setting up the experiment and for the introduction to the particle generators. We would also like to express our sincere thanks to Ralf Käthner for his valuable support with data acquisition troubleshooting.

*Financial support.* The publication of this article was funded by the Open Access Fund of the Leibniz Association.

*Review statement.* This paper was edited by Pierre Herckes and reviewed by four anonymous referees.

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

**Remarks from the language copy-editor**

CE1    Please confirm the change made here.

CE2    Should d be changed to $\Delta$? Please also check this in the figures.

CE3    Should this read "filter-loading" throughout?

CE4    Should this read "BC-loaded" throughout?

CE5    Please confirm the accuracy of the sentence rewording here.

CE6    Please confirm the accuracy of the change made here.

CE7    Please confirm the accuracy of the sentence rewording here.

CE8    Is this to official name of this product? Please check.

CE9    Should there be a value here?

CE10    Please confirm the accuracy of the change made here.

CE11    Please note that Fig. 1 and 3–10 were edited during copy-editing. Please review the figure content carefully.

CE12    Please confirm the accuracy of the sentence rewording here.

CE13    Please check throughout – should this read "areal loading density" (as written elsewhere)?

CE14    Please confirm the accuracy of the change made here.

CE15    Can "in dependency of" be changed to "dependent on" throughout? Please check.

CE16    Please note the slight changes to this section, as edits to this section are not displayed in the track-changes PDF.

CE17    Is this a variable (which should be italicized)?

**Remarks from the typesetter**

TS1    The composition of all figures has been adjusted to our standards.

TS2    Please check spelling. It is "Markowicz" in the bibliography.

TS3    Please check spelling. It is "Cepeda" in the bibliography.

TS4    Please check publication year.

TS5    Please provide this section in full sentences.

TS6    Please check. Reference is not mentioned in the text.

TS7    Please provide date of last access.

TS8    Please check. Reference is not mentioned in the text.

TS9    Please check. Reference is not mentioned in the text.

TS10    Please check. Reference is not mentioned in the text.

TS11    Please add date and place of the conference.

TS12    Please provide date of last access.