# Peer review of "The effect of rapid relative humidity changes on fast filter-based aerosol particle light absorption measurements: uncertainties and correction schemes"

_Atmospheric Measurement Techniques, 2019_

## Referee Comment (RC1) · Anonymous Referee #3 · 23 Jun 2019

The authors present a study of humidity effects on filter-based absorption measurements. These effects may be important with UAV-based (low-power filter-based) absorption measurements where flight durations are relatively short and so the drone may not spend a lot of time at a fixed altitude. These are some interesting experiments, and the authors nicely explain the theory of absorption measurements. I have not seen a lot of papers about the TAP, so this could have been a useful contribution.

However, I have several concerns with this manuscript. It is not clear to me that the authors understand the instruments they use or what the data are telling them. Some

key points are listed below:

1. The TAP has a reference filter measurement, where the particle-free air downstream of the first filter is exposed to a reference measurement. This could explain why the TAP shows a lower response to humidity changes, and possibly why the sign of the effect is opposite that of the mini-Aeth. However, the authors fail to mention this crucial design difference.

2. The effects due to humidity changes are *temporary*. This is shown by the rise in absorption followed by a return to normalcy. The authors try to explain the initial changes by speculations about physical phenomena, but the measurements return to that shown by normal, unaffected conditions even as the RH remains high. The water film or beads would not disappear if the humidity remains constant.

3. A lot of the manuscript focuses on 60-second average measurements. The typical flight time of a UAV is 30 minutes, maybe two hours maximum. At the short end, 60-second averages are not very useful. At the top end, the instrument has time to equilibrate at a particular altitude to wait out any RH effects.

4. The authors present an experiment showing that a dryer reduces the observed RH effects, but don't seem to understand or at least fail to explain why - the dryer likely reduces RH at the filter, so the effect on measured absorption should correspond to that at lower RH. This is seen by the similarity in slopes between the non-dryer and with-dryer cases. A key analysis would be (a) measuring the RH post-dryer; (b) comparing the effect at the post-dryer RH (say 90% pre-dryer, post-dryer 55%) to that at the same non-dryer RH (55%).

5. The authors compare the dry-to-wet and wet-to-dry changes without considering whether these changes are of the same magnitude both ways. There could be hysteresis effects, similar to particle hygroscopicity.

6. The filter loadings and changes in RH considered here are ridiculously high. See

for example doi:10.5194/amt-6-2115-2013; figure attached. RH changes are more gradual and Bap values are much, much lower, which suggests that we will not see the high spikes reported here (except maybe in biomass smoke plumes). The high filter loadings used here (~50 microg/m3) are likely exacerbating any effects.

There are other minor issues, and the manuscript needs a once-over by a native English speaker.

[Figure]

**T. S. Bates et al.: UAS measurements of atmospheric aerosols**

[Figure]

**Fig. 5.** The first 45 min of flight 15 (26 April 2011) showing the initial assent from the airport to 2700 m above the Kongsfjorden. Temperature and humidity (left), total particle number concentration (CN) (middle), and aerosol light absorption coefficient at three wavelengths (right). The mean CN concentration and aerosol light absorption coefficient measured at the Gruvebadet Laboratory during the STADS were 400 cm$^{-3}$ and 0.56 Mm$^{-1}$, respectively.

**Fig. 1.**

---

## Referee Comment (RC2) · Anonymous Referee #4 · 28 Jun 2019

Düsing et al: The effect of rapid relative humidity changes on fast filter-based aerosol particle light absorption measurements: uncertainties and correction schemes, amt-2019-101

**Review**

**General**

The paper describes the effects of variable humidity on measurements of light absorption with filter-based absorption photometers. It has been known for a long time that elevated relative humidity distorts these measurements and this is the main reason why it has been recommended to only use data sampled at dry conditions. In certain type of measurements, especially recently popular balloon or drone-borne vertical measurements drying is not possible and rh is rapidly changing. It is therefore important to characterize the responses of the absorption photometers to the changing conditions. This is what this paper presents. It is probably the first one that actually quantifies the effect.

It is an interesting piece of work. For me the most interesting observation was the completely opposite response of the two absorption photometers and actually of the different filter materials. This very interesting indeed. Actually, it should be emphasized in the conclusions that the responses are filter material dependend and should be quantified if and when new filter materials are used in whatever filter-based absorption photometer, not just the two instruments used in this work. I can recommend the publication of the paper in AMT after some modifications. I did not find any major errors even though I did find some relatively small points to correct and change to the revised version of the ms. I will list them in the detailed comments below

**Detailed comments**

1) L106. "Ogren (2010) published the loading correction ..." and then the Eq. (4) is shown. This is not quite correct. Ogren (2010) presented a corrected version of the equation which was originally presented and also corrected by Bond et al. (1999).

2) L124-126 "... we used the $\sigma_{abs}$ directly provided by the STAP and derived with the mentioned *MAC* in the case for the MA200, which already accounts for multiple scattering and filter loading corrections." How does MA200 account for multiple scattering and filter loading? What function is valid for Teflon? For STAP they are probably assumed to be done with the multiplication by Eq. (4), right?¨

3) The manufacturers, their addresses, and filter materials used in the photometers are presented on lines 79-82, lines 129-132, lines 135-137, and 147-150. Maybe once would be enough.

4) L140. The reference to Holder et al. (2018) is to a conference abstract. I checked it at the conference book of abstracts. Sure, the abstract is there but it is so short that it does not contain any of the information you write on lines 139-145. If you cannot find anything that can be checked by a reader, you should remove these lines.

5) L213-214 "... Filter loading mass is calculated by multiplying the apparent loading mass concentration of the considered material..." What is apparent loading mass? Define it. Where do you get it from?

6) L221-222 " Four different $\rho^*$ were .. for STAP, three for the MA200 ...". Were they not sampling simultaneously?

7) L249. When I look at fig 5 I see that the time when ammonium sulfate was sampled was hours.  How stable could you keep the AS production? How would possible instabilities affect the result?

8) L257-261. There is speculation about possible effects of the negligbly small imaginary index of AS. There is a more plausible explanation. Why wouldn't the explanation be the apparent absorption or cross sensitivity of any filter-based absorption photometer to purely scattering aerosol that the authors are well aware of? The apparent absorption should be mentioned and discussed at some point of the paper already earlier.

9) Section 3.3. This section contains no other information but that a dryer dries humid air.  The points in fig 7 are on the same line with and without drying  so it does not tell anything about the responses of the absorption photometers. You would have obtained the same points also by reducing the original humidity. Even hypothetically there should not be a difference in reducing the original humidity or reducing it afterwards with a drier. Just omit the section.

10) L325-326. Please show a scatter plot of the exponential decay, not only the  time series.

11) Explain clearly in the text and in the figure captions what is the difference
between figs 3 and 5 and figs 4 and 6

---

## Author Comment (AC2) · 20 Sep 2019

We thank for the comments. The remarks are answered in the supplemented pdf.

Please also note the supplement to this comment:
https://www.atmos-meas-tech-discuss.net/amt-2019-101/amt-2019-101-AC2-supplement.pdf
* * *

---

## Author Comment (AC3) · 20 Sep 2019

We thank for the detailed comments. We supplemented the response to the review in the zip file. This also includes the word documents of the new manuscript in the revised version as well as in marked up state to track the changes.

Please also note the supplement to this comment:
https://www.atmos-meas-tech-discuss.net/amt-2019-101/amt-2019-101-AC3-supplement.zip

---

## Author Response (AR1)

**Response to all reviewers.**

**General informations:**

Changes in the manuscript are highlighted with dark red. *Italic* written text is the citation of the reviewers. Blue written text is the response of the authors. Longer text passages additionally included in the manuscript are highlighted with red. The revised version of the manuscript will be uploaded.

**Response to review – RC1**

**Review of anonymous Referee #3**

Remark: „*The authors present a study of humidity effects on filter-based absorption measurements. These effects may be important with UAV-based (low-power filter-based) absorption measurements where flight durations are relatively short and so the drone may not spend a lot of time at a fixed altitude. These are some interesting experiments, and the authors nicely explain the theory of absorption measurements. I have not seen a lot of papers about the TAP, so this could have been a useful contribution. However, I have several concerns with this manuscript. It is not clear to me that the authors understand the instruments they use or what the data are telling them. Some key points are listed below:*"

Response:

We thank for the review. The reviewer focuses on UAV measurements. Though other applications for the STAP are available like in helicopter-borne and tethered balloon platforms. Ongoing we will address the key points of the reviewer point by point:

Key point 1: "*The TAP has a reference filter measurement, where the particle-free air downstream of the first filter is exposed to a reference measurement. This could explain why the TAP shows a lower response to humidity changes, and possibly why the sign of the effect is opposite that of the mini-Aeth. However, the authors fail to mention this crucial design difference.*"

Response:

Thanks for the comment. We totally agree that we forgot to mention that crucial difference in design. To determine the absorption coefficient, the MA200 also uses a reference spot through which no air flow is passing through. Therefore, *rh* changes have a larger impact for this instrument than for the STAP in which the reference spot downstream the sample spot is exposed to RH changes as well. We updated the sections 2.2.1 and 2.2.2 to provide a more detailed look into the design of the instruments. The sections are as follows:

**"2.2.1 Single Channel Tri-color Absorption Photometer (STAP)**

The Single Channel Tri-color Absorption Photometer. This photometer detects light intensities behind two quartz-fiber glass filter (PALL LifeScience, Pallflex Membrane Filters Type E70-2075W) at three wavelengths (450, 525 and 624 nm). On the first filter, the sample filter, the light attenuates due to

deposited particulate matter. The second filter, the reference filter, is located downstream the sample filter and allows blank filter reference light intensity measurements.

By default, the particle light absorption coefficient is determined internally using 60 s averages of the raw intensity measurements for both filter spots. Therefore, in Eq. (5) *I(t)* is defined as:

$$I(t) = \frac{I_{smp,\lambda}}{I_{ref,\lambda}}, \tag{7}$$

where $I_{smp}$ and $I_{ref}$ is the intensity of light at the certain wavelength λ behind the sample (smp) and blank reference (ref) filter, respectively. Nevertheless, all raw measurements are recorded with a time resolution of 1 Hz allowing a recalculation of $\sigma_{abs}$ at this time resolution. The volumetric flow is set to one liter per minute (lpm). According to the manual, at an internal averaging interval of 60 s, the measurement uncertainty is specified to 0.2 Mm$^{-1}$. The spot diameter is ~4.8 mm which leads to a sample area of $A_{spot}$ ~1.75e-5 m$^2$.

**2.2.2 MA200**

The second instrument used here is the microAeth® MA200 is a small sized (13.7 x 8.5 x 3.6 cm; 420g), absorption photometer measuring the attenuation of light at 5 wavelengths (375, 470, 528, 625, and 880 nm; 625 nm are investigated in this study) due to deposited particulate matter on a PTFE filter band.

Similarly to the STAP, the MA200 detects light intensities behind a sample and reference spot. The particulate matter samples on a sample spot with 3 mm diameter leading to a sample area of $A_{spot}$ ~0.71e-5 m$^2$. The reference spot of same area allows for blank filter measurements. $M_{eBC}$ is determined under the assumption that the change of attenuation is proportional to the deposited eBC mass. The measurements were recorded with a 1 Hz time resolution. With the DualSpot® technology the instrument is able to reduce uncertainties related to loading effects up to 60 % (Holder et al., 2018) but was not functioning at the time of the experiment.

Holder et al. (2018) reported that the measurements are slightly depending on *rh* and *T* of the aerosol sample. However, they observed concentrations of up to 7 mg m$^{-3}$, at which the observed dependence on humidity and temperature did not influence the measured values significantly. Furthermore, they used another version of the instrument (MA350), which may react differently to changes in humidity and temperature."

Key point 2: "*The effects due to humidity changes are \*temporary\*. This is shown by the rise in absorption followed by a return to normalcy. The authors try to explain the initial changes by speculations about physical phenomena, but the measurements return to that shown by normal, unaffected conditions even as the RH remains high. The water film or beads would not disappear if the humidity remains constant.*"

**Response:**

Thanks for that comment. We totally agree that the effects are temporary. But, the main focus of this paper is to highlight these temporary changes, since they are important, especially when a high temporal and spatial resolution is needed. The focus of our study is not on measurements with a long averaging time.

Indeed, the beads would not disappear. But after the filter equilibrates after humidity changes there is no change of the optical properties (attenuation) of the filter medium. Since only change of the attenuation of two subsequent measurements is important for the measured particle light coefficient a constant relative humidity has no effect on the measurements.

Key point 3: "*A lot of the manuscript focuses on 60-second average measurements. The typical flight time of a UAV is 30 minutes, maybe two hours maximum. At the short end, 60-second averages are not very useful. At the top end, the instrument has time to equilibrate at a particular altitude to wait out any RH effects.*"

**Response:**

Thanks for that comment. Indeed, for some applications the averaging period is too long. However, the 60-second averaging part of the paper, was intended to qualitatively describe the observed effect. The reaction of the instruments is also shown on a one second time base in Figure 3. These are important to resolve the effect of fast relative humidity changes which is necessary results to develop a correction scheme for the *rh* effect. Equilibration is not intended within this paper, since boundary layer physics, especially in the transition zone of mixing layer and free troposphere, do not allow for equilibration. Also measurements in clouds can be characterized by a fast changing relative humidity. Anyhow, we updated the manuscript by including Figure 3 to show the behavior of both instruments at 1 Hz time resolution in dependency of the *rh* change rate d*rh*/d*t*.

[Figure]

**Figure 1: 1 Hz raw data of $\sigma_{abs}$ at 625 nm measured by the MA200 (blue points) and recalculated at 624 nmSTAP200 (red points), the smooth fit through the measurements (orange and black), and d*rh*/d*t* (purple line).**

Key point 4: "*The authors present an experiment showing that a dryer reduces the observed RH effects, but don't seem to understand or at least fail to explain why - the dryer likely reduces RH at the filter, so the effect on measured absorption should correspond to that at lower RH. This is seen by the similarity in slopes between the non-dryer and with-dryer cases. A key analysis would be (a) measuring the RH post-dryer; (b) comparing the effect at the post-dryer RH (say 90% pre-dryer, post-dryer 55%) to that at the same non-dryer RH (55%).*"

Response:

Thanks for that comment. We refer to anonymous referee #4 (RC2 supplement) that section 3.3 only describes that a dryer dries humidified air and recommends to omit this section. We therefore removed the section from the manuscript.

Key point 5: "*The authors compare the dry-to-wet and wet-to-dry changes without considering whether these changes are of the same magnitude both ways. There could be hysteresis effects, similar to particle hygroscopicity.*"

Response:

Thanks for that comment. We refer here to the scatter plot figures (Fig. 5 and 7 in the new manuscript) which clearly shows the dependency to the sign of the *rh* change. Obviously, the magnitudes are the same for both cases, at least on the basis of the used averaging period which might include a possible hysteresis effect. We clarified within the manuscript and added the sentence "As shown in Fig. 4, the magnitude of the deviation due to positive or negative changes in humidity is approximately the same for each device in terms of magnitude." at the end of Section 3.1. The sentence "Similar to the clean case, for both instruments, drying and humidifying the sample stream to the same extent resulted in a deviation of $\sigma_{abs}$ with the similar magnitude." was added at the end of Section 3.2.1 and at the end of section 3.2.2 we added: "Overall, the magnitude of the deviation of $\sigma_{abs}$ was independent of the sigh of humidity change for both instruments.".

Key point 6: "*The filter loadings and changes in RH considered here are ridiculously high. See for example doi:10.5194/amt-6-2115-2013; figure attached. RH changes are more gradual and Bap values are much, much lower, which suggests that we will not see the high spikes reported here (except maybe in biomass smoke plumes). The high filter loadings used here (~50 microg/m3) are likely exacerbating any effects.*"

Response:

Thanks for that comment. We agree, the filter loading concentrations are high. Anyhow, the loading periods where very short, and the attenuation due to loading with soot was never below 0.52 (extreme case), but the majority was above 0.74 which is a reasonable value for multi used filters or measurements in polluted conditions. Additionally, the filter loading concentrations cannot exacerbate the observed Considering *rh* change rates, we agree, especially in the case for 1 second average periods the change rates were pretty large. Anyhow, we also presented cases with change rates of -1.42 to 1.09 % s$^{-1}$ which can be observed under real atmospheric conditions. Furthermore, since the observed points lying on the same line with a constant slope over all considered *rh* changes it is a satisfying assumption that the slope is also

valid for very small *rh* changes resulting in smaller bias. To point out that the filters were loaded before the humidified sample airstream was collected by the instruments we added: "Two main setups were used to investigate the effect of changes in real humidity. In the first, the filters of the devices were unloaded and the instruments collected a particle free airflow with adjustable relative humidity to examine the pure filter effect. In the second, the filters of the devices are loaded to a certain degree and afterwards they sample particle-free humidified air which accounts for the combination of both effects, the pure filter effect and the effect induced by the hygroscopic behavior of the particles." as second last paragraph in Section 2.3.

General point 1: "*There are other minor issues, and the manuscript needs an once-over by a native English speaker.*"

Response:

Thanks for that comment. We read over the manuscript and changed some unclear sentences. We refer here to the supplemented revised version of the manuscript in which all changes are documented.

**Response to review – RC2**

**Review of anonymous Referee #4**

Remark: „ *The paper describes the effects of variable humidity on measurements of light absorption with filter-based absorption photometers. It has been known for a long time that elevated relative humidity distorts these measurements and this is the main reason why it has been recommended to only use data sampled at dry conditions. In certain type of measurements, especially recently popular balloon or drone-borne vertical measurements drying is not possible and rh is rapidly changing. It is therefore important to characterize the responses of the absorption photometers to the changing conditions. This is what this paper presents. It is probably the first one that actually quantifies the effect.*

*It is an interesting piece of work. For me the most interesting observation was the completely opposite response of the two absorption photometers and actually of the different filter materials. This very interesting indeed. Actually, it should be emphasized in the conclusions that the responses are filter material dependend and should be quantified if and when new filter materials are used in whatever filter-based absorption photometer, not just the two instruments used in this work. I can recommend the publication of the paper in AMT after some modifications. I did not find any major errors even though I did find some relatively small points to correct and change to the revised version of the ms. I will list them in the detailed comments below:*"

**Response:**

We thank for the review. We will consider the general remark and emphasize, that it is a crucial task to quantify the effect of any absorption photometer, either by following our experimental setup or by other similar approaches. Especially since the STAP can operate with different filter material. Furthermore, more measurements regarding the MA200 have to be conducted to fully understand all the processes. In the conclusion is already stated that the opposing behavior is caused by the different filter material. As recommendation we added: "5. Since the response is different in magnitude and sign for both filter materials, we recommend to examine the effect for other filter materials as well."

The reviewer provided more comments and ongoing we will address them point by point:

Key point 1: *"L106. "Ogren (2010) published the loading correction ..." and then the Eq. (4) is shown. This is not quite correct. Ogren (2010) presented a corrected version of the equation which was originally presented and also corrected by Bond et al. (1999)."*

Response:

Thanks for that comment. We think the reviewer means that the sentence itself is wrong, and we update to: "For instance, Ogren (2010) reported an updated loading correction function for the PSAP introduced and updated by Bond et al. (1999) defined as:". We directly took the Eq. from the manual of the STAP so the Eq. (4) should be mentioned, since it is used internally.

Key point 2: *"L124-126 "... we used the $\sigma_{abs}$ directly provided by the STAP and derived with the mentioned MAC in the case for the MA200, which already accounts for multiple scattering and filter loading corrections." How does MA200 account for multiple scattering and filter loading? What function is valid for Teflon? For STAP they are probably assumed to be done with the multiplication by Eq. (4), right?"*

Response:

Thanks for that comment. Yes, for the STAP the correction function from Eq. (4) is used internally by the STAP itself to account for filter loading effects. In the manual of the MA200 there is no loading correction function given. Anyhow, Jimenez et al. (2007) empirically determined a loading correction function *K(ATN)* for the Teflon filter-based AE41 and AE42 (Magee Scientific Company). This equation corrects the measured black carbon mass concentration $M_{BC}$ as follows: $M_{BC} = \dfrac{b_{ATN}}{\sigma_{ATN}} \times \dfrac{1}{K(ATN)}$,

in which $b_{ATN}$ is the attenuation coefficient in $m^{-1}$ and $\sigma_{ATN}$ the attenuation efficiency of BC with a value of 16.6 $m^2\,g^{-1}$ at 880 nm. *K(ATN)* is defined as:

$$K(ATN) = a + b \times \exp(\frac{-ATN}{100}),$$

with *a* and *b* some linear regression coefficients. Jimenez at al. (2007) reported values for *a* and *b* of 0.13 and 0.88. Besides that correction function, which could be applied during post-processing, the MA200 features a DualSpot© loading correction approach. Another Aethalometer (AE33) also uses the dual spot approach. A comprehensive correction approach using light attenuation measurements of two sample spots is provided by Drinovec et al. (2015). But, since the DualSpot© mode was not working during the time of the experiment, this also cannot be applied to the given data set. Hence, we suspect that there was no loading correction applied internally to the measurements of the MA200. Furthermore, since we do not know, if the MA200 needs the same loading correction of the AE41 or AE42, we did not consider any loading correction and decided to use the data directly measured by the instrument. This, if so, should only influence the results by a view per cent. As shown in the given manuscript, the areal loading density as well as the loading material do not change the response of the MA200 to *rh* changes. Therefore, we think a loading correction is negligible.

References:  Jorge Jimenez, Candis Claiborn, Timothy Larson, Timothy Gould, ThomasW. Kirchstetter & Lara Gundel (2007) Loading Effect Correction for Real-Time AethalometerMeasurements of Fresh Diesel

Soot, Journal of the Air & Waste Management Association, 57:7,868-873, DOI: 10.3155/1047-3289.57.7.868

Drinovec, L., Močnik, G., Zotter, P., Prévôt, A. S. H., Ruckstuhl, C., Coz, E., Rupakheti, M., Sciare, J., Müller, T., Wiedensohler, A., and Hansen, A. D. A.: The "dual-spot" Aethalometer: an improved measurement of aerosol black carbon with real-time loading compensation, Atmos. Meas. Tech., 8, 1965-1979, https://doi.org/10.5194/amt-8-1965-2015, 2015.

Key point 3: *"2) The manufacturers, their addresses, and filter materials used in the photometers are presented on lines 79-82, lines 129-132, lines 135-137, and 147-150. Maybe once would be enough."*

Response:

Thanks for that comment. Yes, we totally agree, that mentioning once is enough. Therefore, we removed the first, second and fourth mentioning and kept the manufactures addresses in the instrument description part. The used filter materials are important, therefore we think it is worth to repeat the used filter material several times to emphasize that.

Key point 4: *"L140. The reference to Holder et al. (2018) is to a conference abstract. I checked it at the conference book of abstracts. Sure, the abstract is there but it is so short that it does not contain any of the information you write on lines 139-145. If you cannot find anything that can be checked by a reader, you should remove these lines."*

Response:

Thanks for that comment. The given reference refers to a poster presented during the 10[th] IAC in St. Louis, MO, USA. We updated the reference. We included the url which links to the webpage where the poster can be downloaded. This contains all the given information mentioned in the manuscript.

Holder, A., B. Seay, A. Brashear, T. Yelverton, J. Blair, and S. Blair. Evaluation of a multi-wavelength black carbon sensor, Poster, 10th International Aerosol Conference, St. Louis, MO, September 02 - 07, url: https://cfpub.epa.gov/si/si_public_record_report.cfm?Lab=NRMRL&dirEntryId=342614, 2018.

Key point 5: *"L213-214 "... Filter loading mass is calculated by multiplying the apparent loading mass concentration of the considered material..." What is apparent loading mass? Define it. Where do you get it from?"*

Response:

Thanks for that comment. We think we did not used the correct words to explain what we meant. We therefor reworded the 5[th] sentence in Section 3.2.: "For both considered loading materials, the mass loaded onto the filters was calculated by multiplying the prevalent loading mass concentration within the mixing chamber with the volume flow rate of the instrument and the loading duration.". The calculation of the prevalent loading mass concentration within the chamber is explained for both loading materials in the beginning of the Section 3.2.1 and 3.2.2 separately. For ammonium sulfate the ammonium sulfate

volume concentration (integral of the volume size distribution) within the chamber was multiplied an assumed ammonium sulfate density of 1.77 g/cm$^{-3}$. For BC different approaches had to be considered. We added: "During experiment #1 the mean absorption coefficient of the STAP was divided by a *MAC* of 6.6 m$^2$ g$^{-1}$ since the absorption was stable during the loading period and it's a direct measure from the sampling instrument. For the experiment #2 the loading mass concentration was taken from the average of two consecutive MAAP measurements since the loading period was shorter than 2 minutes which is shorter than the internal averaging period of the STAP so that no stable absorption coefficient readouts could be provided by the STAP. During experiment #3 no MAAP was available and the absorption coefficient measured by the STAP was unstable. We therefore decided to estimate the loaded eBC mass by integrating the absorption coefficient during the loading period and dividing it by the *MAC*.". This should clarify how the loadings on the filters were derived. Furthermore, we think that the word "apparent" is wrong it that context.

Key point 6: "*L221-222 " Four different ρ\* were .. for STAP, three for the MA200 ...". Were they not sampling simultaneously?*"

Response:

Thanks for that comment. For both instruments the first three filter areal loading densities were estimated simultaneously. Because of the lower flow rate and the different spot sizes these densities are different. The fourth loading density in the case of the BC was only estimated for the STAP since no MA200 was available during this time. Anyhow, the loading density is in the same range of previous loading periods and therefore at least on case with that areal loading density is covered. The MA200 was also not available during another ammonium sulfate experiment and therefore we have considered 11 areal loading densities for the STAP and 8 for the MA200 in this case. We changed the third last sentence in Section 2.3 to: "The loading aerosol was split into two streams from one of which the absorption photometer were sampling simultaneously." to emphasize that both photometer were sampling the same aerosol.

Key point 7: "*When I look at fig 5 I see that the time when ammonium sulfate was sampled was hours. How stable could you keep the AS production? How would possible instabilities affect the result?*"

Response:

Thanks for that comment. We consider, that we missed to explicitly point out that in the experiments considering different areal loading densities, the filter was loaded before the *rh* was changed. We therefore added the sentence:" The filter were loaded to a certain extent with different materials and afterwards the absorption photometer were sampling particle free air with adjustable humidity." In Section 3.2. Also we added:" Two main setups were used to investigate the effect of changes in real humidity. In the first the filters of the devices were unloaded and the instruments collected a particle free airflow with adjustable relative humidity. In the second, the filters of the devices are loaded to a certain degree and afterwards they sample particle-free humidified air." as the second last paragraph in Section 2.3.

The loading periods lasted 20 minutes at most. During the loading periods we generated ammonium sulfate with an atomizer, which produces very stable loading mass concentrations (narrow standard

deviation around the mean particle number and volume size distribution in Figure 2) when the aerosol chamber is well-mixed. Only during the well-mixed states of the mixing-chamber the filters were loaded. We added: "The very narrow standard deviation around the mean particle number and volume size distribution in Figure 2 indicate clearly that the loading mass concentrations were very stable during the loading periods." at the end of the first paragraph in Section 3.2.2.

Key point 8: *"L257-261. There is speculation about possible effects of the negligibly small imaginary index of AS. There is a more plausible explanation. Why wouldn't the explanation be the apparent absorption or cross sensitivity of any filter-based absorption photometer to purely scattering aerosol that the authors are well aware of? The apparent absorption should be mentioned and discussed at some point of the paper already earlier."*

Response:

Thanks for that comment. This part is speculative and we removed the very negligibly small imaginary part of ammonium sulfate as a possible explanation. We agree that the cross-sensitivity of the absorption photometer to purely scattering aerosol is more important. The problem is, that the sensitivity to ammonium sulfate (Fig 6., measuring absorption during the loading period around 18:30 and 21:00 UTC) is also visible for the MA200 and it is not showing any variation in the response to relative humidity changes across different loading materials which could mean that PTFE membrane filter is unaffected by filter loading in terms of the *rh* effect or the loading was too low. This has to be tested in further studys. Anyhow, we updated the 5th paragraph in Section 3.2.2 to: "As shown in Fig. 6, both absorption photometers measure an "apparent" absorption coefficient of approximately 2 Mm$^{-1}$ during loading with ammonium sulfate (18:30 and 21:00 UTC). This shows that absorption photometers react sensitively to scattering aerosols such as ammonium sulfate. The scattering ability of any material can be described with the real part of its refractive index. It seems that for the STAP the slope of the correlation increases with increasing scattering of the loading material (0.15 Mm$^{-1}$ %$^{-1}$ for a clean filter, 0.21 Mm$^{-1}$ %$^{-1}$ for ammonium sulfate, and 0.30 Mm$^{-1}$ %$^{-1}$ for BC). Ammonium sulfate has a real part of 1.521 ± 0.002 (at 532 nm Dinar et al, 2007) and BC from combustion processes has a real part of 1.96 at 530 nm (Kim et al., 2015 following Ackermann and Toon (1981)). Hence, the quartz fiber glass filters loaded with "artificially" absorbing aerosol inside the STAP could lead to a variation in the response to relative humidity changes. But, the MA200 was loaded with ammonium sulfate as well and its response to relative humidity changes is almost constant for all considered loading materials. Therefore, either the observation is caused by the interaction of quartz fiber glass filters with the loading material and the PTFE filter inside the MA200 do not causes this behavior, the filter loading of the MA200 was too low, or there are other mechanisms explaining this. Furthermore, since only three different cases (clean, ammonium sulfate and BC) were observed in this study more materials should be considered to investigate this phenomenon.".

Key point 9: *"Section 3.3. This section contains no other information but that a dryer dries humid air. The points in fig 7 are on the same line with and without drying so it does not tell anything about the responses of the absorption photometers. You would have obtained the same points also by reducing the original humidity. Even hypothetically there should not be a difference in reducing the original humidity or reducing it afterwards with a drier. Just omit the section."*

**Response**:

Thanks for that comment. You are right. We omitted the Section since no new findings are given.

Key point 10: *"L325-326. Please show a scatter plot of the exponential decay, not only the time series."*

**Response**:

Thanks for that comment. We are not quite sure what you mean with showing the exponential decay with a scatter plot in particular. Showing the decay for all investigated points is not helpful since the 1 Hz data is a) very noisy and b) the magnitude depends on the *rh* change and is c) biased due to the response time of the *rh* sensor. Therefore we added a new Figure (Figure 3) displaying the exponential decay exemplarily for *rh* change periods at 14:22 UTC and 14:32 UTC (*rh* change from Fig. 4 at the same time). We added:"

[Figure]

**Figure 3: 1 Hz raw data of $\sigma_{abs}$ at 625 nm measured by the MA200 (blue points) and STAP200 (red points), the smooth fit through the measurements (orange and black) and d*rh*/d*t* (purple line)."**

and referenced to this Fig. within the text : "The exponential recovery behavior of the MA200 (see Figure 1) requires a more complex approach to correct for relative humidity changes." (first sentence ins Sect. 3.3.2) and also we mentioned the exponential recovery behavior in the beginning of Sect. 3: "This chapter will give an overview of the measurement results. The overall behavior of both instruments will be shown for wavelengths of 624 nm in the case of the STAP and 625 nm in the case of the MA200, respectively. A closer look at the behavior of both devices at 1 Hz time resolution shows that both devices differ greatly in quality (see Figure 3). The STAP (red dots and the smooth fit shown as black line) reacts very fast to

relative humidity changes (d$rh$/d$t$ as purple line) and then returns relatively fast to the zero line. The MA200, on the other hand, also shows a fast response to relative humidity changes, but then shows a distinct exponential recovery (see Figure 3, blue dots and smooth fit shown as orange line) and reports absorption coefficients although there is no $rh$ change.". The smooth fit (orange line) clearly indicates an exponential recovery behavior.

Key point 11: "*Explain clearly in the text and in the figure captions what is the difference between figs 3 and 5 and figs 4 and 6.*"

Response:

Thanks for that comment. Caption of Figure 4 is now: "Figure 4: Time series of $rh$ (top panel) and absorption coefficient (bottom panel) measured with STAP (624 nm; black) and MA200 (625 nm; red) with clean filters." and of Figure 6: "Figure 6: $rh$ of the air stream sampled by the MA200 and the STAP (upper panel) and $\sigma_{abs}$ measured by MA200 and STAP at 625 (624) nm (lower panel). First up and down ramp of $rh$ conducted with clean filter, second and third under conditions with filter loaded with ammonium sulfate. Loading periods around 18:30 and 21:00 UTC.". Caption of Figure 5 states now: " Figure 5: Scatter plot (dots) of all observations of the absolute excursion of $\sigma_{abs}$ ($\Delta\sigma_{abs}$) in dependence of the absolute change in $rh$ ($\Delta rh$), its linear regression fit as well as the summarizing boxplot of the linear regression fit are shown for the three investigated states (clean, loaded with BC and ammonium sulfate) at 624 nm (STAP, black colors) and 625 nm (MA200, red colors). Descriptive coefficients are given in Appendixtable 1." and correspondingly shows Fig. 7 the same plot but only considers the maximum and minimum areal loading density of the respective loading material. Caption of Fig. 7 states now: " Figure 7: Scatter plot of change in absorption ($\Delta\sigma_{abs}$) in dependence of the absolute change in $rh$ ($\Delta rh$) separated into the different loading states (loaded with BC and ammonium sulfate) and minimum and maximum loading areal density on the filter. Dashed and colored lines represent the linear regression fit. Red and blue colors indicate MA200 at 625 nm and black and green colors indicate STAP at 624 nm. In the first panel BC loading is shown whereas in the second panel the ammonium sulfate case is displayed. Coefficients of the linear regression fit are displayed in panel 3. Shading of color in the linear fits and of the points are same as in panel 3."

The main difference between Fig. 4 and Fig. 6 is that in Fig. 6 smaller, stepwise $rh$ changes were conducted. We updated the paragraph 3 in Section 3.2.2 to: "Figure 6 shows exemplarily the time series of $rh$ of the sampled air $rh$ and of the $\sigma_{abs}$ measured with STAP and MA200 operated with clean filters. A $rh$ of 0.0 to 96.2% with $drh/dt$ humidity change rates of in the range of -1.42 and to 1.09 % s$^{-1}$ was measured. Compared to the case in Figure 3 here a step-wise change of $rh$ is shown. These steps resulted in a smaller absolute excursion of $\sigma_{abs}$ which ranges from -7.2 to 9.0 Mm$^{-1}$ (STAP; 624 nm, 60 s measurement resolution) and -14.1 to 10.9 Mm$^{-1}$ (MA200; 625 nm, 60 second running mean). Furthermore, Figure 6 shows the response of the $\sigma_{abs}$ to $rh$ changes at three different states of filter loading. During the first ramp the filter were clean, during the second period the filters had a filter areal loading density of 32.5 (STAP) and 12.4 (MA200) mg m$^{-2}$ and during the third ramp the filter in the STAP had loading areal density of 98.7 mg m$^{-2}$ and the MA200 filter was loaded with an areal loading density of 37.6 mg m$^{-2}$. The response of the instruments during these periods is shown in Appendixtable 1."

The difference between Figure 4 and Figure 6 is that Fig. 4 shows the overall (mean) behavior of both instruments in BC and Ammonium sulfate case whereas Fig. 6 shows the response behavior of both instruments in cases of minimum and maximum loading of BC and Ammonium sulfate. The first sentence

in paragraph 4 Section 3.2.2 already states now: "In Figure 4 (lower left panel), the overall (mean) response of both instruments to *rh* changes is shown in the case of loading with ammonium sulfate." The second last sentence in the last paragraph of Section 3.2.2 already stated: "In Figure 6 (middle panel), the spread of the slopes within the shown cases is exemplarily shown for the investigated minimum and maximum load of the filters.".

**Response to review – RC3**

**Review of anonymous Referee #1**

Remark: „*Düesing et al. have provided a systematic and detailed characterization of the bias of two widely-used commercial absorption photometers which results from exposure to step RH changes. While they have not solved the problems of these photometers, they have nevertheless provided useful quantitative data and useful correction schemes. The manuscript should be published in ACP after addressing all of my comments below. The most important comments are that the data must be weighted by uncertainties before fitting, that the running mean the authors used has smoothed the data (and likely biased the fits), and that complete uncertainties must be provided for the authors' correction schemes.*"

**Response:**

We thank for the review. We will consider the general remark. Regarding the weighting of the fittings. The *rh* sensor uncertainty is constant in absolute value of 1.8% up to 90% *rh*. But the sensor includes also a response time which has to be considered as well. This is a very complex task and we cannot estimate the uncertainty induced by that. Therefore a weighting by uncertainties is not possible. The reviewer provided detailed comments and ongoing we will address them point by point:

Key point 1: "*Section 2.1 (Theory of instruments) should be expanded to include mathematical statements of how the authors view the transient RH effects. In particular, it should be spelled out that $M_{eBC}$ is based on the difference between subsequent attenuation measurements. This differential attenuation measurement also raises the possibility of investigating and correcting RH effects by looking directly at attenuation data. The authors should either look into this possibility, or discuss why they did not.*"

**Response:**

Thanks for that comment. We did not consider the attenuation because for most of the users of this instruments the particle light absorption coefficient and the eBC mass concentration are more intuitive.

Considering the effect of water in form of water vapor expressed as relative humidity we changed after Eq. (5): " *Water has a refractive index of 1.33+i1.5e-9 at 532 nm wavelength. Hence it interacts with incoming electromagnetic radiation. If the filter is exposed to a relative humidity changes the light attenuation of the filter changes simultaneously, since the water binds to the filter itself (Caroll, 1976 and Caroll, 1986). Since a variety of filter materials, with different physical properties exist, we suspect that magnitude and sign of the light attenuation coefficient can vary with the filter material. The hypothesis is that the change rate of the rh (drh/dt) directly determines the magnitude of the particle light absorption coefficient, which depends on the difference of two subsequent attenuation measurements.*". We think that clarifies the paragraph.

Key point 2: *"The use of a running mean for the MA200 means that the results are not equivalent to the 1-minute mean of the STAP. The running mean approach needs to be reconsidered. First, a running 60-second mean results in smoothing since each data point is used 3 times. Therefore the linear fits and R2 values reported are invalid since R2 is artificially enhanced by the autocorrelation which is inherent in a running mean. Best practice would be to analyze the 1-sec MA200 data and 60-sec mean MA200 data. The difference will provide insight into the STAP's limited time resolution. This point is related to my next point."*

and

Key point 3: *"The changes in Figure 3 are rapid relative to the 1-minute averaging intervals used. This means that the signal cannot be accurately represented by a single value (mean) during periods of change (increasing/decreasing RH). I would predict that increasing/decreasing RH periods have systematically different biases in the residuals of Figure 4. To correctly account for these biases, uncertainties must be estimated and an orthogonal regression must be performed in Figure 4, after weighting by these uncertainties. Most scientific software packages support this. Afterwards please highlight periods of increasing/decreasing RH in Figure 4 (eg with different symbols)."*

**Response to point 2 and 3:**

Thanks for that comments. We choose the given averaging periods to show the general behavior of both instruments. For correction, the 1 Hz data of instruments were used. The linear regression is not artificially enhanced since we selected discrete points in the time series for the correlation. We described the method in the beginning of Section 3: *"This chapter will give an overview of the measurement results. The overall behavior of both instruments will be shown for wavelengths of 624 nm in the case of the STAP and 625 nm in the case of the MA200, respectively. A closer look at the behavior of both devices at 1 Hz time resolution shows that both devices differ greatly in quality. The STAP reacts very fast to relative humidity changes (see Figure 3, red dots and orange line) and then returns relatively fast to the zero line. The MA200, on the other hand, also shows a fast response to relative humidity changes, but then shows a distinct exponential recovery (see Figure 3) and reports absorption coefficients different from zero although there is no rh change.*

*Therefore, we use an averaging on a 60 second basis to describe the qualitative behavior of both devices. In the case of the STAP, the internal 60 second averaging is used. For the MA200, on the other hand, a 60-second "running average" is applied to the 1 Hz measurements.*

*The qualitative behavior of both devices is shown as follows. To each absolute change in the relative humidity (Δrh) the corresponding maximum of the excursion of the averaged absorption coefficient (Δσ$_{abs}$) has been assigned. Where the absolute change of the relative humidity is the difference between the relative humidity at the time of the largest excursion in the absorption coefficient and the relative humidity at the start of the excursion. This approach also excludes the response time of the rh sensor.*

*First, the results for the pure filter effect will be shown. Afterwards, we present the results of the combined behavior of filter and aerosol particles on the filters. For loaded filters, the combined effect will be shown separated into BC and ammonium sulfate loaded filter."*

Furthermore, the STAP has no limited time resolution since it reports the raw intensities with a 1Hz resolution and from which the light absorption coefficient can be calculated based on Eq. (5). Anyhow, the different behavior of both instruments on a 1 Hz measurement resolution (exponential recovery of the MA200, see Fig. 3) led to the conclusion that we apply a running mean to the data of the MA200, which is

similar to the internal running average of the STAP to describe the qualitative behavior of both instruments on that time base. Also, we included, that the *rh* sensor has a response time ($t_{63}$) of less than 10 s (see sensor specifications) as the 2$^{nd}$ last sentence in paragraph 3 of Sect. 2.3:" *Furthermore, this sensor has a response time $t_{63}$ of <10s.*". The correlation of discrete points of the time series also accounts for that response time.

The scatter plot in Fig. 4 (now Fig. 5) shows the absolute change of absorption in dependency of the absolute change of *rh*. Therefore, the last non-excursed value (before *rh* change) of the absorption coefficient was subtracted from the maximum value of the absorption coefficient during the excursion. This was assigned to the difference of the starting *rh* and the *rh* at the point of maximum excursion of $\sigma_{abs}$. To clarify we included the description of the method as shown above. We furthermore think, that it was unclear that we did not correlate the whole time series of *rh* and $\sigma_{abs}$ but the difference of discrete points in the time series. We think the updates make this clearer. Therefore, the points in Fig. 4 (now Fig. 5) do not need another symbol since points estimated based on a positive *rh* change are located in the right half of the scatter plots.

Key point 4: "*The authors have speculated extensively about the cause of the opposite trends of quartz and PTFE (lines 197-200). This speculation is of little value without experimental support. But I am not requesting experimental support. I am rather suggesting that the authors use these insights to design an improvement — use a mixture of the MA200 and TAP approaches to cancel out some of the biases of each approach. The utility of this suggestion can be tested by "simulating" a new instrument using the authors' measurements. The design details related to feasibility of this should be commented on.*"

**Response:**

Thanks for that comment. Since the underlying processes, especially the exponential recovery behavior of the MA200, are not fully understood. A new instrument could contain two sampling spots with both filters to cancel out each other. For that the different magnitudes of the effects have to be considered as well. It is only speculative why the PTFE filter inside the MA200 shows a larger response to *rh* changes. Besides the flow not passing through the reference spot, also the lower flow rate could have an effect but was not investigated. Therefore, we think a "simulation" of a new instrument is not useful right now. Anyhow, we included a paragraph, which gives a first idea how such a new instrument could be designed: "Since the filter in the STAP reveals a positive and the filter in the MA200 a negative correlation to relative humidity changes a combination of both filters within one instrument could account for the observed effect. A new developed instrument could use these two different filter materials on two sampling spots to cancel out the effect of each other. Though, more investigations have to be done, especially to understand the different recovery behaviors and effect magnitudes of the PTFE and quartz-fiber filter.".

Key point 5: *"In Figure 7, why did the authors not simply sample for a longer time with the MA200 in order to match the loadings on either instrument?"*

Response:

Thanks for that comment. The loading of the STAP and MA200 under the given loading conditions can be converted to equivalent sampling periods of several hours depending on the prevalent ambient aerosol mass concentration. We wanted to keep these equivalent sampling equal for both instruments. Furthermore, under real life conditions the MA200 samples less particulate matter than the STAP by default due to the smaller flow rates.

Key point 6: *"Line 260, not only the imaginary part of refractive index but also the real part will affect these results, since the real part will influence scattering (influencing attenuation as well as subsequent absorption). Please reword."*

Response:

Thanks for that comment. We overthought this and came to the conclusion that the imaginary part does not has an impact on the shown behavior since it is too small in the case of ammonium sulfate. We reworded also following the comments of anonymous referee #4 and we updated the 5[th] paragraph in Section 3.2.2 to: "As shown in Fig. 5, both absorption photometers measure an "apparent" absorption coefficient of approximately 2 $Mm^{-1}$ during loading with ammonium sulfate (18:30 and 21:00 UTC). This shows that absorption photometers react sensitively to scattering aerosols such as ammonium sulfate. The scattering ability of any material can be described with the real part of its refractive index. It seems that for the STAP the slope of the correlation increases with increasing scattering of the loading material (0.15 $Mm^{-1}$ $\%^{-1}$ for a clean filter, 0.21 $Mm^{-1}$ $\%^{-1}$ for ammonium sulfate, and 0.30 $Mm^{-1}$ $\%^{-1}$ for BC). Ammonium sulfate has a real part of 1.521 ± 0.002 (at 532 nm Dinar et al, 2007) and BC from combustion processes has a real part of 1.96 at 530 nm (Kim et al., 2015 following Ackermann and Toon (1981)). Hence, the quartz fiber glass filters loaded with "artificially" absorbing aerosol inside the STAP could lead to a variation in the response to relative humidity changes. But, the MA200 was loaded with ammonium sulfate as well and its response to relative humidity changes is almost constant for all considered loading materials. Therefore, either the observation is caused by the interaction of quartz fiber glass filters with the loading material and the PTFE filter inside the MA200 do not causes this behavior, the filter loading of the MA200 was too low, or there are other mechanisms explaining this. Furthermore, since only three different cases (clean, ammonium sulfate and BC) were observed in this study more materials should be considered to investigate this phenomenon."

Key point 8: "*The correction schemes are not perfect, but they are useful. Certainly these and other authors will apply them at some point. It is therefore very important to report UNCERTAINTIES for the correction schemes. Both a percentage uncertainty and a bias (absolute value, in analogy to limit of detection) uncertainty must be reported. The bias requirement is illustrated in Figure 10, where 2/Mm of false signal result from a step RH change of about 30%. This bias of 2/Mm means that a true signal of 1/Mm would hardly be measurable.*

*I do not know of a formal reference for handling this kind of bias, but I have encountered it in my own work and thought a bit about an easily understandable solution. My best suggestion is to allow users to answer the question: what is the minimum reported value which I can trust, if I am willing to accept a maximum inaccuracy of 25%? This question can be answered with a simple mathematical formulation which I will leave for the authors to provide. The answer to this question (the actual bias) will obviously depend on the magnitude of Δrh.*"

**Response:**

Thanks for that comment. We reworded a major fraction of Section 3.3.1 including the correction scheme of the STAP. It states now:" In Figure 8 the correlation of $rh$ change rate ($drh/dt$) and the measured $\sigma_{abs}$ at 624 nm measured by the STAP (red circles) and recalculated with respect to standard conditions (pressure of 1013.25 hPa and temperature 273.15 K) is shown. The STAP-based background eBC mass concentration during the experiment was ~190 ng m$^3$ (at standard conditions, $\sigma_{abs}$ at 624 nm converted with a *MAC* of 6.6 m$^2$ g$^{-1}$), which corresponds to offset (standard conditions corrected values) in the shown scatterplot of Figure 8 and which has no influence on the response to $rh$ changes as shown previously.

The $rh$ change rate ranged from -10.8 to 14.5 % s$^{-1}$. These rates correspond to a $\sigma_{abs}$ of -231 to 192 Mm$^{-1}$ for recalculated values at standard conditions and -203 to 164 Mm$^{-1}$ directly measured by the instrument. But these measurements are biased by the response time of the relative humidity sensor so that the "real" $rh$ change-rate cannot fully represented by these measurements. On average the slope (correction factor $C_{rh}$ in Eq. (8)) of the linear fit is 10.08 (± 0.12) Mm$^{-1}$ s %$^{-1}$ for standard conditions and 8.82 (± 0.10) Mm$^{-1}$ s %$^{-1}$ for direct instrument output, respectively. Calculating the particle light absorption coefficient introduced by $rh$ changes with:

$$\sigma_{abs,rh} = C_{rh} \frac{drh}{dt} \qquad (8)$$

for different $rh$ change rates in both, the recalculated and direct instrument output case, and subtracting it from measurements allows to correct for the observed effect as follows:

$$\sigma_{abs,corr} = \sigma_{abs,meas} - \sigma_{abs,rh}, \qquad (9)$$

and after replacing $\sigma_{abs,rh}$ in Eq. (8) with Eq. (9) follows:

$$\sigma_{abs,corr} = \sigma_{abs,meas} - C_{rh} \frac{drh}{dt}. \qquad (10)$$

The y-intersect of the linear fit in Figure 8 has not to be considered for correction as mentioned before. Disadvantageously, with this correction the noise of the $rh$ sensor will propagate in the corrected $\sigma_{abs}$.

Furthermore, the linear fit in Figure 8 under- or overestimates the behavior in regimes of very high relative humidity change rates most likely due to the response time of the *rh* sensor, so that the correction function cannot entirely correct the bias. Therefore, the given correction factor $C_{rh}$ consists of uncertainties, which cannot be entirely addressed. Hence, it is only a first guess, needs further refinement and right now we do not recommend to use the correction approach as long the uncertainties are not fully addressed. Furthermore, since only one STAP was tested, other STAP may have other correction factors due to a unit to unit variability. Additionally, other filter materials used in the STAP can also lead to another behavior. Anyhow, the upper function was applied to STAP measurements conducted with the same *rh* sensor under atmospheric conditions.

Exemplarily, Figure 9 shows this application. The figure shows airborne measurements of $\sigma_{abs}$ at 624 nm derived with the STAP derived during a campaign conducted in March 2017 in East Germany. The upper panel displays the *rh* of a dried aerosol sample stream measured upstream of the STAP. The lower panel shows the recalculated $\sigma_{abs}$ at 624 nm wavelength corrected for *rh* changes (black) and biased by *rh* changes (red). In the periods where the *rh* changes relatively fast (d*rh*/d*t* of -0.55 to 0.56 % s$^{-1}$ e.g. at around 6200 seconds), the uncorrected $\sigma_{abs}$ overshoots. The correction significantly reduces this bias and smooth out the measurements during the periods of *rh* changes. At the peaks of d*rh*/d*t* the difference of the corrected and uncorrected values is up to 1.5 Mm$^{-1}$, which is significant with respect to the measured $\sigma_{abs}$. The periods with negative $\sigma_{abs}$ are not introduced by the *rh* effect. We moreover think that a small offset is introduced in the initialization process of the instrument. Despite the imperfection of the correction scheme, this linear approach can be useful to derive a rough estimate of the accuracy of the measurements. For instance let *x* be the required accuracy for the measurements in % and $\sigma_{abs}$ the measured particle light absorption coefficient we can express the ambient particle light absorption coefficient which is at least needed to fulfill the accuracy criterion in dependency of the *rh* change rate d*rh*/d*t*:

$$\sigma_{abs,meas} \geq \frac{100\%}{x[\%]} C_{rh} \left[ \frac{Mm^{-1}s}{\%} \right] \left| \frac{drh}{dt} \right| \left[ \frac{\%}{s} \right]. \tag{11}$$

Exemplarily, if a change rate of 0.1 % s$^{-1}$ is measured and an accuracy of 25% is needed, at least a measured particle light absorption coefficient of around 4 Mm$^{-1}$ is needed to fulfill the accuracy criterion.".

The main reason why we did not provide any uncertainties of the correction scheme besides the uncertainty of the slope is that we simply cannot quantify the uncertainty introduced by the response time of the *rh* sensor. Furthermore, each *rh* sensor will have different characteristics so that the correction scheme, if any, can only be applied using this sensor. We moreover suggest to use the findings to estimate the measurement uncertainties introduced by *rh* changes and to set a lower threshold of reliable measurements depending on the required accuracy and prevalent *rh* change rate.

For the MA200, the problem is even more complex since the correction approach results in slightly different coefficients of the correction formula when applied to other similar experiments. This could be due to a unit-to-unit variability or other phenomena affecting the PTFE response so that not all uncertainties can be addressed. Furthermore, the response time of the *rh* sensor introduces some uncertainty to the correction approach which cannot be quantified. Since, we do not recommend to use the correction scheme providing uncertainties is of little value. Similar to the STAP the correction approach looks a) promising and shows the right direction and b) could be used to roughly estimate the bias in the measurements due to *rh* changes. Also, the last sentence of the second last paragraph and the last

paragraph of Section 3.3.2 states now: *"Here, the response time of the sensor could account at least for a part of the imperfection of the correction approach and cannot be fully quantified, yet.*

*Unfortunately, the application of the same correction approach to other similar experiments resulted in different correction function a and b. Applying the approach to two clean case experiments from section 3.1 resulted in optimized parameters of a = -0.92 and -1.03 and b = 0.974 and 0.971, respectively. Hence, it is just a first step trying to account for relative humidity changes and further research with more MA200 simultaneously has to be done to fully understand the underlying processes and to fully quantify the uncertainties of the correction scheme Nevertheless, the presented approach significantly reduces the amplitude of the bias in the shown data set (see Figure 10). But, up to now we cannot recommend to use the given parameters to correct for rh effects. At most it can be used to make a rough estimate of how measurements of the particle light absorption coefficient derived the MA200 could be biased by rh changes."*.

Minor comments:

We thank the reviewer for all the minor points, we will comment on each separately point by point.

Point 1: *"1. I would suggest taking the natural logarithm of Equations 1 and 2, or at least 2, so that the important terms (exponents of e) are more easily visible. Also, please at line 99 add a sentence clarifying that reinterpreting l as an aerosol path length does not mean that σ represents the aerosol absorption coefficient but still the filter attenuation coefficient."*

Response:

We expressed Eq. (1) and Eq. (2) following the reviewer. In principle the theory should explain how the particle light absorption coefficient is derived. We followed the recommendations of the reviewer and added the sentence: "But, a reinterpretation of the path length does not mean that the result is the particle light absorption coefficient, but still the light attenuation coefficient.". But, we used the term light attenuation coefficient instead of filter attenuation coefficient.

Point 2: *"Line 116, please change "provide" to "report" since the photometers only estimate eBC."*

Response:

We changed according the referees comment.

Point 3: *"Comparison" by who, are those unpublished results from the authors' lab?*

Response:

Yes, these are unpublished results from the authors' lab and the data can be requested if needed. We changed the last paragraph in Sect. 2.1 to: "*Lab-comparison of the eBC mass concentration between a MAAP (Multi Angle Absorption Photometer; Thermo Fisher Scientific, 27 Forge Parkway, 02038 Franklin, MA, USA; Petzold and Schönlinner, 2004) at 637 nm wavelength and MA200 at 625 nm and STAP at 624*

*nm beforehand the experiment revealed a good agreement within 3% and within 6%, respectively. For the STAP a MAC of 6.6 $m^2 g^{-1}$ was assumed. Since a MAC of 6.6 $m^2 g^{-1}$ is used for the MAAP at 637 nm, in this study we used the $\sigma_{abs}$ directly provided by the STAP and derived with the mentioned MAC in the case for the MA200, which already accounts for multiple scattering and filter loading corrections."*

Point 4: "*Line 138 and 155, I suggest SI units of area*".

**Response:**

Thanks for the comment. We are not sure what the reviewer means with Si-units of area. To our opinion $m^2$ is already a SI-unit.

Point 5:" *Line 167, change "by passing" to "by passing it through" (this sentence required 3 reads to be understood)*"

**Response:**

We updated to: "One of the flows was humidified by passing through two glass tubes containing distilled water at room temperature with an inlet and outlet for compressed particle free air.".

Point 6: "*I have not seen the term "floating mean" used before and an internet search did not bring up any definitions. I would recommend "running mean" (more precise, since floating implies complete freedom whereas running implies autocorrelation).*"

**Response:**

We changed each occurrence of floating mean with running mean.

Point 7: *The Section "Recommendations" should be a numbered section or subsection, and no sections should come after Conclusions.*

and

Point 8: *In Recommendations and the Introduction, the authors suggest avoiding fast changes by ascending slowly. This is simply not possible in some scenarios (unmixed layers, clouds) and this should be noted.*

**Response to point 7 and 8:**

Thanks for the comments. We removed the heading of the recommendations section and changed to the paragraph with the recommendations to: "The findings summarized above lead to following recommendations how to use this type of instruments:

1. When used for vertical profiling, apparent sharp gradients in *rh* during the profile have to be taken into account.
   a. The ascending speed of the profiling platform should be reduced if possible, to decrease the temporal change of *rh*, but in some scenarios this is simply not possible and therefore,

- b. when fast relative humidity changes cannot be avoided, such periods have to be removed from the data set, or at least to estimate the uncertainties of the measurements based on the presented correction functions. Therefore,
2. we recommend recording the *rh* of the sampled aerosol. This allows to determine *rh* change rates. This allows to roughly estimate the bias of *rh* changes on filter-based absorption measurements with these two instruments.
3. The usage of a dryer is highly recommended, because it reduces the amplitude of the excursion in the measurements during fast *rh* changes.
4. For both instruments we recommend to conduct more similar experiments to address the flaws of our study to refine the presented correction approaches.
5. Since the response is different in magnitude and sign for both filter materials, we recommend to examine the effect for other filter materials as well.".

Also, we added in the last paragraph of the Abstract: "*Due to our findings, we recommend to use an aerosol dryer upstream of absorption photometers to reduce the rh effect significantly. Furthermore, when absorption photometers are used in vertical measurements, the ascending or descending speed through layers of large rh gradients has to be low to minimize the observed rh effect. But this is simply not possible in some scenarios especially in unmixed layers or clouds. Additionally, recording the rh of the sample stream allows correcting for the bias during post processing of the data. This data correction leads to reasonable results, according the given example in this study.*".

Point 9: *"Table 1: I see no bold entries."* and point 10*: "Table 2: Instead of custom formatting, add a column "Filter Number" which increases by 1 when appropriate."*

**Response:**

Thanks for the comment. We updated the tables to:"

**Table 1: Filter loading mass concentration ($M_{eBC}$) of the black carbon particles and filter areal loading density (deposited mass per spot area) $\rho^*_i$. $M_{eBC}$ were determined by dividing the average $\sigma_{abs}$ of the STAP with an assumed MAC of 6.6 m$^2$ g$^{-1}$ or based on the MAAP measurements. Usage of same filter is indicated by its filter number. Bold written entries were used for the investigation of the *rh* effect.**

| filter number | $M_{eBC}$ [μg m$^{-3}$] | $\rho^*_{eBC,i}$ [mg m$^{-2}$] | |
|---|---|---|---|
| | | STAP | MA200 |
| #1 | **44.5 (STAP)** | **14.0** | **5.4** |
| | 43.4 (STAP) | 37.9 | 14.4 |
| | **27.6 (STAP)** | **42.9** | **16.3** |
| #2 | **52.6 (MAAP, 2 scans)** | **2.8** | **1.1** |
| #3 | - | **13.7 (integral of STAP)** | no data |

**Table 2: Average volume and mass concentration ($V_{(NH4)2SO4}$, $M_{(NH4)2SO4}$) of the loading (NH₄)₂SO₄ aerosol derived from the used MPSS (number of used scans in brackets) and loading areal density $\rho^*_{(NH4)2SO4}$ of the filters are given. Usage of same filter is indicated by its filter number, which means that the filter loading mass was adding up during the experiments.**

| filter number | $V_{(NH4)2SO4}$ [µm³ cm⁻³] (# scans) | $M_{(NH4)2SO4}$ [µg m⁻³] | $\rho^*_{(NH4)2SO4}$ [mg m⁻²] | |
|---|---|---|---|---|
| | | | STAP | MA200 |
| #1 | 15.4 (2) | 27.2 | 3.1 | 1.2 |
| | 18.6 (1) | 32.9 | 10.5 | 4.0 |
| | 20.6 (3) | 36.4 | 31.3 | 11.9 |
| #2 | 20.6 (4) | 36.5 | 40.8 | 15.5 |
| #3 | 33.1 (3) | 58.6 | 32.5 | 12.4 |
| | 33.5 (5) | 59.3 | 98.7 | 37.6 |
| #4 | 20.3 (3) | 36.0 | 21.1 | 8.0 |
| | 20.3 (3) | 36.0 | 41.9 | 15.9 |
| #5 | 23.9 (3) | 42.4 | 28.9 | no data |
| | 28.4 (4) | 50.2 | 69.8 | no data |
| | 29.8 (2) | 52.8 | 99.6 | no data |

The changes within the manuscript are manifold and we refer here to the marked-up version below, which contains all the changes.

The new Figure 3 is included in the supplemental material with all the other figures.

[revised manuscript text omitted]